# Individual characteristics outperform resting-state fMRI for the prediction of behavioral phenotypes
Amir Omidvarnia [1,2] ✉, Leonard Sasse [1,2,3], Daouia I. Larabi [4], Federico Raimondo [1,2], Felix Hoffstaedter [1,2], Jan Kasper[1,2], Jürgen Dukart[1,2], Marvin Petersen [5], Bastian Cheng [5], Götz Thomalla[5], Simon B. Eickhoff[1,2] & Kaustubh R. Patil [1,2]

In this study, we aimed to compare imaging-based features of brain function, measured by resting-state fMRI (rsfMRI), with individual characteristics such as age, gender, and total intracranial volume to predict behavioral measures. We developed a machine learning framework based on rsfMRI features in a dataset of 20,000 healthy individuals from the UK Biobank, focusing on temporal complexity and functional connectivity measures. Our analysis across four behavioral phenotypes revealed that both temporal complexity and functional connectivity measures provide comparable predictive performance. However, individual characteristics consistently outperformed rsfMRI features in predictive accuracy, particularly in analyses involving smaller sample sizes. Integrating rsfMRI features with demographic data sometimes enhanced predictive outcomes. The efficacy of different predictive modeling techniques and the choice of brain parcellation atlas were also examined, showing no significant influence on the results. To summarize, while individual characteristics are superior to rsfMRI in predicting behavioral phenotypes, rsfMRI still conveys additional predictive value in the context of machine learning, such as investigating the role of specific brain regions in behavioral phenotypes.

Resting-state functional magnetic resonance imaging (rsfMRI) is a widely used neuroimaging modality for studying human brain function[1–3]. Functional connectivity is an important aspect of rsfMRI defined as the statistical dependence between different brain areas during periods of rest or low cognitive demand[4]. A common application of rsfMRI is the prediction of cognitive performance[5–8], and clinical phenotypes[7,9–12]. A common approach is to extract multiple features from rsfMRI and utilize them in predictive modeling. This method has been boosted by modern MRI scanners with high magnetic field strengths, big public datasets, high-performance computing systems, computationally enhanced software packages, and more efficient machine learning algorithms[13]. However, the field has struggled to advance to real-world applications due to systematic challenges such as modest prediction accuracy in large populations ($N_{subject} > 2000$) and replication failures of studies with small sample sizes[7,8,14,15]. An effective improvement could be to consider features beyond prevalent functional connectivity measures.

In many of the current rsfMRI-based prediction pipelines, individual characteristics such as age, gender, and total intracranial volume (TIV) are typically treated as confounds and hence removed from the rsfMRI features or from the prediction targets[16]. The rationale behind this practice is that any information other than that directly related to brain activity should be discarded because it may prevent us from determining the neuronal origin of the predictive signal[17]. However, there is evidence that features based on individual characteristics might be better at predicting measures of mental health than those based on fMRI[18,19]. The relative efficacy of rsfMRI and individual characteristics in predicting behavioral phenotypes is still debatable. The capacities and constraints of rsfMRI in behavioral prediction can be better understood by addressing this debate. To investigate this, we developed four behavioral prediction scenarios using a wide range of rsfMRI features in conjunction with three typically considered confounds, age, gender, and TIV. We used a large sample from the UK Biobank ($N_{subject} = 20,000$) that included rsfMRI and four behavioral phenotypes:

[1]Institute of Neuroscience and Medicine, Brain & Behavior (INM-7), Research Center Jülich, Wilhelm-Johnen-Straße, Jülich 52428, Germany. [2]Institute of Systems Neuroscience, Medical Faculty, Heinrich Heine University Düsseldorf, Moorenstr. 5, Düsseldorf 40225, Germany. [3]Max Planck School of Cognition, Stephanstrasse 1a, Leipzig, Germany. [4]Department of Clinical and Developmental Neuropsychology, University of Groningen, Grote Kruisstraat 2/1, 9712 TS Groningen, the Netherlands. [5]Klinik und Poliklinik für Neurologie, Kopf- und Neurozentrum, University Medical Center Hamburg-Eppendorf, Hamburg, Germany. ✉e-mail: a.omidvarnia@fz-juelich.de

fluid intelligence, processing speed, visual memory, and numerical memory[20]. The behavioral phenotypes were then predicted using different rsfMRI features at the brain region of interest (ROI) level.

Functional connectivity and large-scale nonlinear interactions between brain regions are intertwined[21,22]. The nonlinearity of functional connections at micro-, meso-, and macro-scales gives rise to a temporally complex behavior in the hemodynamic response of the brain across time, as measured by fMRI[23,24]. There is evidence that the temporal complexity of rsfMRI and behavioral phenotypes are correlated, providing a promising feature space for brain-behavior predictions[25–29]. As temporal complexity may collect different rsfMRI properties than functional connectivity, it is expected to be able to supplement functional connectivity's prediction capacity. On the other hand, it is unclear how these rsfMRI-derived properties relate to the noise profile of various brain regions, originated from hardware, head movement, heartbeat, and respiration[30]. In the behavioral prediction pipelines of this study, we utilized nine rsfMRI features covering five prominent characteristics of functional connectivity (fractional amplitude of low-frequency fluctuations or $fALFF$[31], local correlation or $LCOR$[32], global correlation or $GCOR$[33], Eigenvector centrality or $EC$[34], and weighted clustering coefficient or $wCC$[34]) as well as four temporal complexity metrics (Hurst exponent or $HE$[35], Weighted permutation entropy or $wPE$[36], Range entropy (type B) or $RangeEn_B$[37], and Multiscale entropy or $MSE$[38]). We then entered these features into the predictive modeling pipelines, considering different roles for age, gender, and TIV as follows: (*i*) rsfMRI features without removing age, gender, and TIV, (*ii*) rsfMRI features after removing of these individual characteristics (i.e., treating them as confounds), (*iii*) a combination of rsfMRI features with age, gender, and TIV, and (*iv*) age, gender, and TIV only. We also investigated the impact of ROI-wise temporal signal to noise ratio (tSNR) on the models to examine the influence of noisy brain regions in the predictions. Figure 1a illustrates the block diagram of the study.

We found that the temporal complexity and functional connectivity features both give about the same predictive power when applied to four different behavioral phenotypes. Nevertheless, individual characteristics routinely outperformed rsfMRI features in predictive accuracy in analyses involving smaller sample sizes. Age, gender, and TIV, when combined with rsfMRI features, improved predictive outcomes. The results were relatively unaffected by the selection of brain atlas or the effectiveness of two predictive modeling methods. The removal of age, gender, and TIV from the features or targets resulted in reduced performance. The results also showed that age and gender could be predicted more accurately than behavioral phenotypes in general. Our findings indicate that individual characteristics outperform rsfMRI in predicting behavioral phenotypes, but rsfMRI can still offer supplementary predictive capability.

## Results
### Statistics and Reproducibility
In order to investigate the generalizability of our findings across brain atlas configurations and predictive models, we performed all analyses using Schaefer400 and Glasser360 brain parcellation atlases as well as linear ridge regression and linear support vector machine (SVM). The results of four atlas-model combinations were consistent across various sample sizes and predictive modeling scenarios (see Fig. 1a). Age, gender, and TIV consistently outperformed rsfMRI features in predicting behavioral phenotypes, irrespective of the atlas and model used. The accuracy curves for predicting behavioral phenotypes over different population sizes using the Schaefer400 atlas were comparable with those obtained using the Glasser360 atlas, for both the ridge regression model (Fig. 2 and Fig. S1) and the linear SVM model (Fig. 2, Fig. S3, S7–S10). We also investigated the impact of tSNR thresholding on the prediction accuracies using the Schaefer400 atlas (Fig. 3 as well as Fig. S18 to Fig. S21). A similar situation was observed when performing age and gender prediction as shown in the pairs of Fig. 4 versus Fig. S4, Fig. S5 versus Fig. S6, Fig. S11 versus Fig. S12, and Fig. S13 versus Fig. S14. This consistent observation suggests that the superior predictive capacity of demographics and TIV as well as their

combinations extend beyond the specific choice of brain atlas and predictive model.

We chose to use ridge regression in our primary study due to its widely recognized application in neuroimaging and behavioral research[39,40]. Fig. S15 demonstrates the histogram of the best alpha parameters which led to the best optimization for the ridge regression models across different scenarios. We then incorporated linear SVM with heuristic hyper parametrization in addition to ridge regression and repeated all analyses using this model. Although both models showed comparable performance at large population sizes, ridge regression demonstrated greater stability, particularly in situations with smaller samples and lower prediction accuracy. This can be observed by comparing the prediction accuracies of visual memory and numeric memory in Fig. S2 with their corresponding curves in Fig. S3. Furthermore, the linear SVM model necessitates a larger population size to achieve its maximum predictive capability, as opposed to the ridge model. This can be observed by comparing the accuracy curves for processing speed prediction in Fig. S2 with the corresponding curves in Fig. S3. It justifies the utilization of ridge regression and classification for the behavioral phenotype prediction when using rsfMRI features.

### Quantifying rsfMRI complex dynamics and behavioral phenotypes
We used preprocessed rsfMRI data from 20,000 unrelated UK Biobank participants[17] and extracted four temporal complexity measures as well as five functional connectivity-derived measures from them (see Methods). As prediction targets, we chose the four most reliable behavioral phenotypes in the UK Biobank database, fluid intelligence, processing speed, visual memory, and numerical memory[41]. See Table S1 in the Supplementary Materials for more details. We used ridge regression with $L2$-norm regularization and linear SVM with heuristic hyper parametrization for predictive modeling, widely used models for behavioral phenotypic prediction using rsfMRI[7,40,42]. Model performance was measured through cross-validation using the Spearman correlation between the real and predicted targets in regression tasks or the balanced accuracy in classification tasks. We used nested cross-validation where the hyper parameter $\alpha$ of ridge regression was tuned in the inner loop[19]. The impact of age, gender, and TIV, was addressed through four scenarios, outlined in Fig. 1 (see also Methods).

### Larger sample sizes increase accuracy but eventually reach a plateau
First, we examined whether increasing the sample size could improve the prediction accuracy of behavioral and non-behavioral targets in all four scenarios (Fig. 1). Increasing the number of subjects improved accuracy most of the time, but the performance curves reached a plateau when including approximately more than 2,000 participants in the analyses (Fig. 2, and Fig. S1 to Fig. S6). As a sanity check, we tested all the predictive modeling scenarios using fish consumption (the day prior to fMRI data collection—data field 103140) as a target presumably unrelated to the rsfMRI features. The performance for all sample sizes, rsfMRI features, individual characteristics, and their combinations remained at chance level (Fig. 2, and Fig. S1–S6). The findings presented in Fig. 2, along with Fig. S1–S3, utilized all brain regions without applying any tSNR thresholding. This included a total of 400 and 360 regions for the Schaefer and Glasser atlases, respectively. Then, we only used the Schaefer400 brain atlas to investigate the impact of tSNR thresholding on the prediction accuracies (Fig. 3 as well as Fig S18–S21). The brain feature maps were subsequently subjected to a thresholding process based on the ROI-wise tSNR values of parcellated rsfMRI datasets. The subset of ROIs that passed the threshold were retained as features for prediction analysis.

### Temporal complexity and functional connectivity features show comparable predictive capacities
Next, we investigated how temporal complexity and functional connectivity features compare in behavioral phenotype prediction across different sample sizes. The average performance of ridge regression and linear SVM

# An overview of the methodological steps in this study

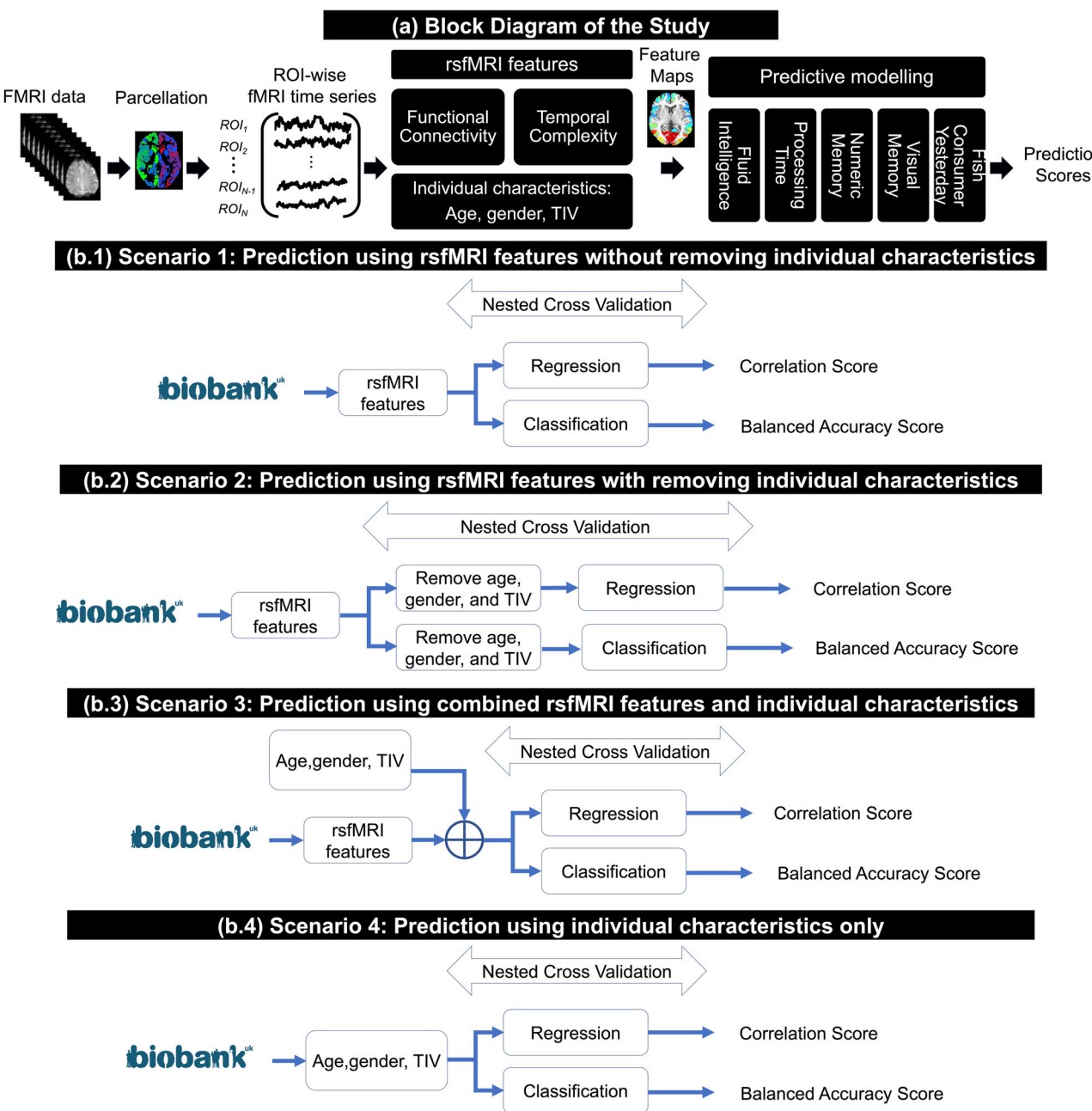

**Fig. 1 | The analysis pipeline and different analysis scenarios of this study. a** Main block diagram of this study, including the rsfMRI features and the prediction targets from the UK Biobank. **b** Four analysis scenarios based on the role of individual characteristics, i.e., age, gender, and total intracranial volume, in behavioral phenotypic prediction. Abbreviations: rsfMRI resting state functional magnetic resonance imaging, TIV total intracranial volume, ROI region of interest.

models across 5 repeats and 5 folds of nested cross-validation suggested that certain features, specifically *fALFF*, *LCOR*, *wPE*, and the AUC of *RangeEn_B*, performed better than others in all contexts, regardless of the target. The correlation between actual and predicted values remained below 0.35 even at the maximum sample size (20,000 individuals). Even with this large sample size, not all behavioral phenotypes could be predicted with equal accuracy (Fig. 2). Fluid intelligence (data field 20016) was predicted with the highest correlation coefficient of up to 0.14, followed by processing speed (data field 20023) and numeric memory (data field 20240) with ~0.1 when using *LCOR* after removing age, gender, and TIV. The prediction accuracy of processing speed was higher than that of the other three behavioral phenotypes when using age, gender, and TIV only (Scenario 4, see Fig. 1). However, as shown

in the black-colored curves of Fig. 2 and the corresponding Supplementary Figs., the predictability of fluid intelligence, visual memory, and numerical memory scores was close to each other.

## Age, gender, and TIV result in higher accuracy than rsfMRI features

Next, we tested how age, gender, and TIV predict behavioral performance when used as sole input features and without any rsfMRI data involved (Fig. 1(b.1)). As shown in Fig. 2 and Fig. S1 to Fig. S3, this scenario resulted in the highest correlation between actual and predicted targets using both ridge regression and SVM modelings across all sample sizes, outperforming all scenarios where rsfMRI features were utilized (Fig. 1(b.1–b.3)). It was also

# Prediction accuracy of ridge modeling - Schaefer400 brain atlas

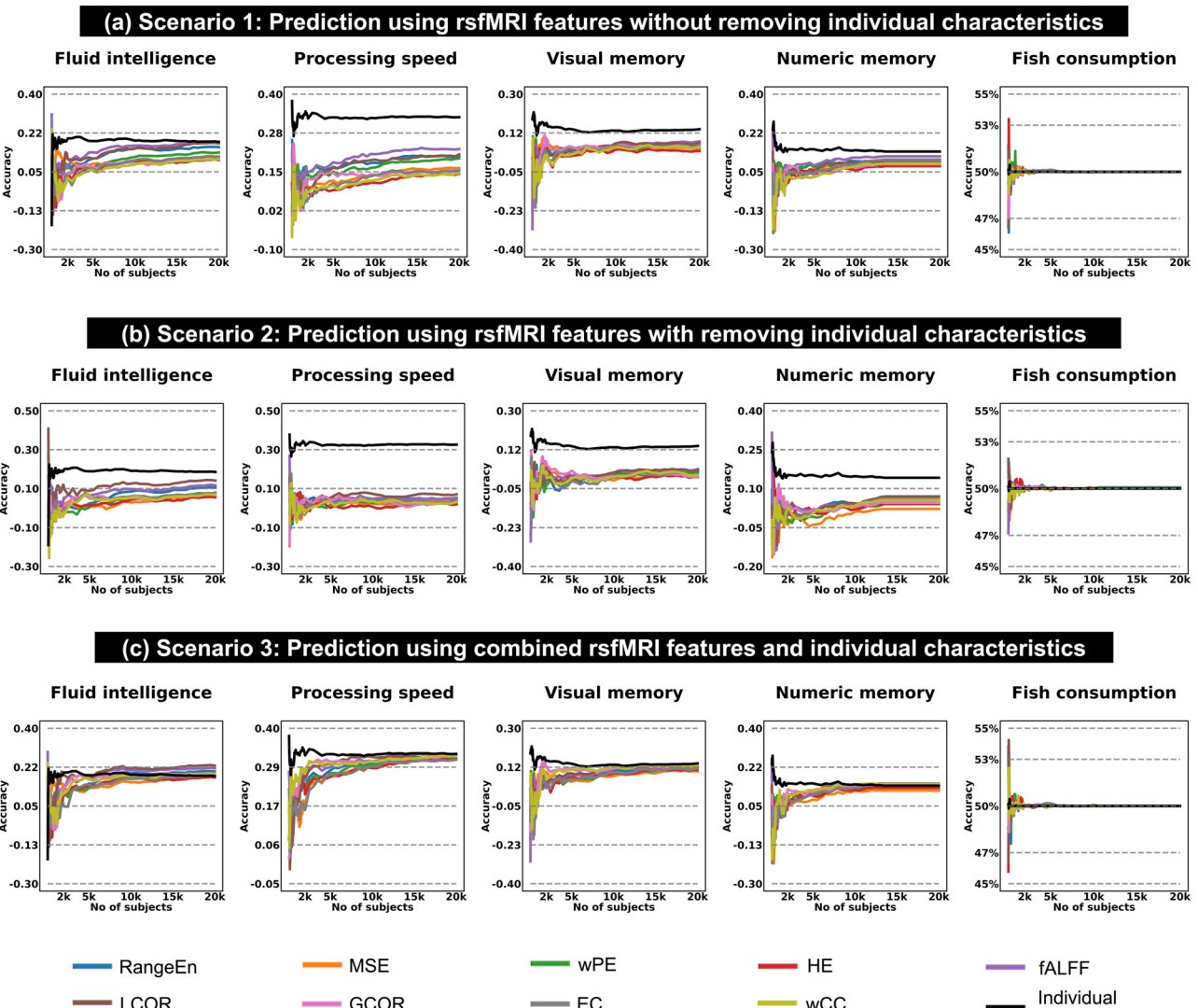

**Fig. 2 | Prediction of different behavioral phenotypes.** The prediction was done across different sample sizes using the Schaefer400 brain atlas and ridge predictive modeling. The x-axis represents the population size in the analysis, ranging from 100 to 20,000 UK Biobank participants. The y-axis shows the prediction accuracy measured by the Spearman correlation coefficient for behavioral phenotypes (fluid intelligence, processing speed, visual memory, and numeric memory) as well as age, and by the balanced accuracy for gender and fish consumption prediction. The prediction accuracy curves for each behavioral phenotype and individual characteristic are color-coded. rsfMRI resting state functional magnetic resonance imaging, PE permutation entropy, fALFF fractional amplitude of low frequency fluctuations, LCOR local correlation, GCOR global correlation, EC eigenvector centrality, CC clustering coefficient.

independent from the brain parcellation atlas used for feature extraction. When individual characteristics served as input features, the sample size required to reach the plateau was also substantially lower (less than 500 subjects). In other words, the ability of individual characteristics to predict behavioral phenotypes from a small sample size was better than the ability of rsfMRI features to predict the same targets, even when a larger sample size was used.

Given that the individual characteristics outperformed rsfMRI features in predicting behavioral phenotypes, the next logical step was to combine each temporal complexity and functional connectivity features with individual characteristics and see if it improves the predictions. For all rsfMRI features, this scenario produced the highest prediction accuracy of the first three analysis scenarios. The distinction between combined rsfMRI features and individual characteristics (Fig. 2, Scenario 3) and rsfMRI features only (Scenarios 1 and 2) was more pronounced when predicting processing speed in comparison to the other three behavioral phenotypes.

Figure S7 to Fig. S10 illustrate the box plot presentations of Fig. 2 and Fig. S1 to Fig. S3.

## Temporal SNR plays no major role

We then asked whether excluding brain regions with low tSNR levels would increase prediction accuracy. We used a group-level tSNR map to threshold the rsfMRI feature maps (see Methods). Figure 3 shows fluid intelligence prediction accuracies for Scenarios 1 to 3 after stepwise thresholding on the tSNR maps from 0% (no threshold, equivalent with the results illustrated in Fig. 2) to 100% (no ROI for prediction) with 5% increments. Each panel in the figure represents a distinct pair of features and targets, with color-coded accuracy values. The x-axis indicates the population size in the analysis, while the y-axis denotes the count of suprathreshold ROIs after tSNR thresholding. The predictive modeling for each feature-target pair was conducted across various sample sizes, spanning $N_{subject} = 100$ to $N_{subject} = 20,000$. For population sizes between 100 and 2000, increments of

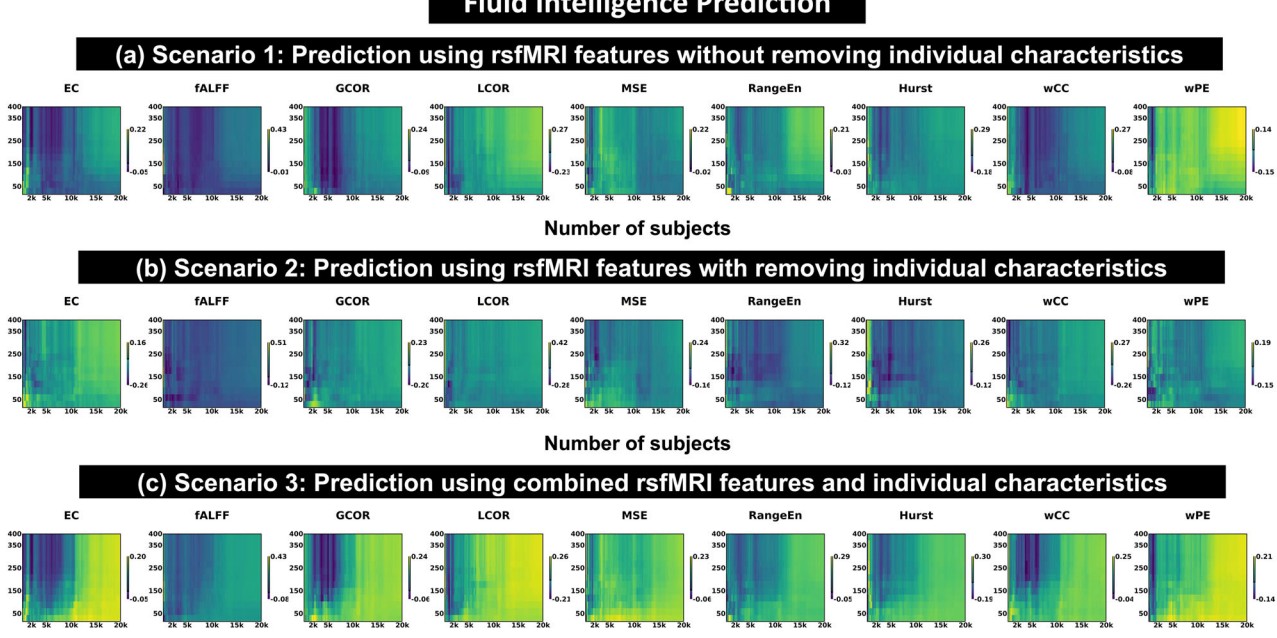

**Fig. 3 | Analysis of tSNR for fluid intelligence prediction.** Spearman correlation accuracy was used with ridge regression modeling of fluid intelligence using nine rsfMRI features and after tSNR thresholding from 0% to 100% through Scenario 1 (**a**), Scenario 2 (**b**), and Scenario 3 (**c**). Schaefer400 brain atlas was used for rsfMRI parcellation. The numbers of suprathreshold ROIs at tSNR threshold level spanning from 0% to 60% with 5% increment for the Schaefer400 and Glasser360 brain atlases are [400, 397, 397, 396, 387, 366, 333, 276, 201, 130, 68, 33, 13] and [360, 360, 356, 353, 338, 311, 270, 203, 142, 83, 57, 25, 10], respectively. For both brain atlases, tSNR levels above 60% led to no suprathreshold ROIs. Abbreviations: rsfMRI resting state functional magnetic resonance imaging, ROI region of interest, tSNR temporal signal to noise ratio, PE permutation entropy, MSE multiscale entropy, RangeEn range entropy, fALFF fractional amplitude of low frequency fluctuations, LCOR local correlation, GCOR global correlation, EC eigenvector centrality, wCC weighted clustering coefficient.

50 subjects were employed, while increments of 500 subjects were applied for the range of 2000 to 20,000. Prediction accuracies improved with increasing sample size and the number of suprathreshold ROIs. The finding was consistent across the other behavioral phenotypes (see Figs. S18–S20). Prediction accuracy for fish consumption remained at chance-level at all tSNR thresholds (Fig. S21). As shown in Fig. 3, we observed that overall, prediction performance improved with the number of suprathreshold ROIs and the tSNR level. However, the figure does not explain which of these two factors are the main driver here. Figure S16 and Fig. S17 show the prediction accuracies as a function of these two factors using the Schaefer400 brain atlas and the linear SVM model with heuristic $C$ for all rsfMRI features and prediction targets at $N_{subject}$ = 20,000. As Fig. S16 illustrates, the prediction accuracies strongly depend on the number of ROIs when this number is low (typically less than 150). However, they increase and become largely independent of the number of ROIs when this number is sufficiently high. Given that the two sets of prediction accuracy curves (shown in Fig. S16 and Fig. S17) show a similar pattern of prediction performance, one can conclude that the ROI count is the main factor influencing the accuracy of the predictions, regardless of the tSNR of the involved ROIs in the prediction procedure. Applying a tSNR threshold of 60% resulted in 13 suprathreshold ROIs using the Schaefer400 brain atlas. This was the minimum number of suprathreshold ROIs that could be detected in the parcellated data following tSNR thresholding. No suprathreshold ROIs were produced when the ROI thresholds were less than 60% of the maximum tSNR across all regions. Figure S17 shows that most of the prediction accuracy curves drop sharply at the tSNR level of about 65%. This means that the feature vectors that were thresholded above this level were not informative enough for prediction because there were not enough suprathreshold ROIs for that. However, the prediction accuracy curves have been presented in the figures from 0% to

100% of tSNR thresholding for the sake of completeness. Figure S22 illustrates the spatial distribution of the tSNR values across brain regions for both brain parcellation atlases. According to Fig. S22, the numbers of suprathreshold ROIs at tSNR threshold level spanning from 0% to 60% with 5% increment for the Schaefer400 and Glasser360 brain atlases are [400, 397, 397, 396, 387, 366, 333, 276, 201, 130, 68, 33, 13] and [360, 360, 356, 353, 338, 311, 270, 203, 142, 83, 57, 25, 10], respectively. For both brain atlases, tSNR levels above 60% led to no suprathreshold ROIs. We have included the nifti files of normalized tSNR maps for both brain atlases in the supplement allowing the readers to directly examine any desired threshold.

**Age and gender are easier to predict than behavioral phenotypes**
We investigated the capability of the rsfMRI features to predict age and gender using ridge regression and linear SVM. *wPE* and *RangeEn_B* (temporal complexity) performed best at large sample sizes, as well as *fALFF* and *LCOR* (functional connectivity), with correlation coefficients of up to 0.5 between the true and predicted targets. This accuracy was considerably better than the prediction accuracy of behavioral phenotypes, which was typically less than 0.25 (see Fig. 2 versus Fig. 4, and Fig. S4 to Fig. S6). This result was noticeably different when individual characteristics were used as features for predictive modeling (gender and TIV for age prediction, and age and TIV for gender prediction). Gender could be classified using age and TIV with more than 85% accuracy. However, gender and TIV did not perform well in age prediction ($\rho = 0.2$). Figure S11 to Fig. S14 illustrate the box plot presentations of Fig. 4 and Figs. S4–S6.

**Similar individual patterns across rsfMRI features**
We observed that some rsfMRI features have comparable predictive capacity, despite their mathematical definitions and interpretations being quite

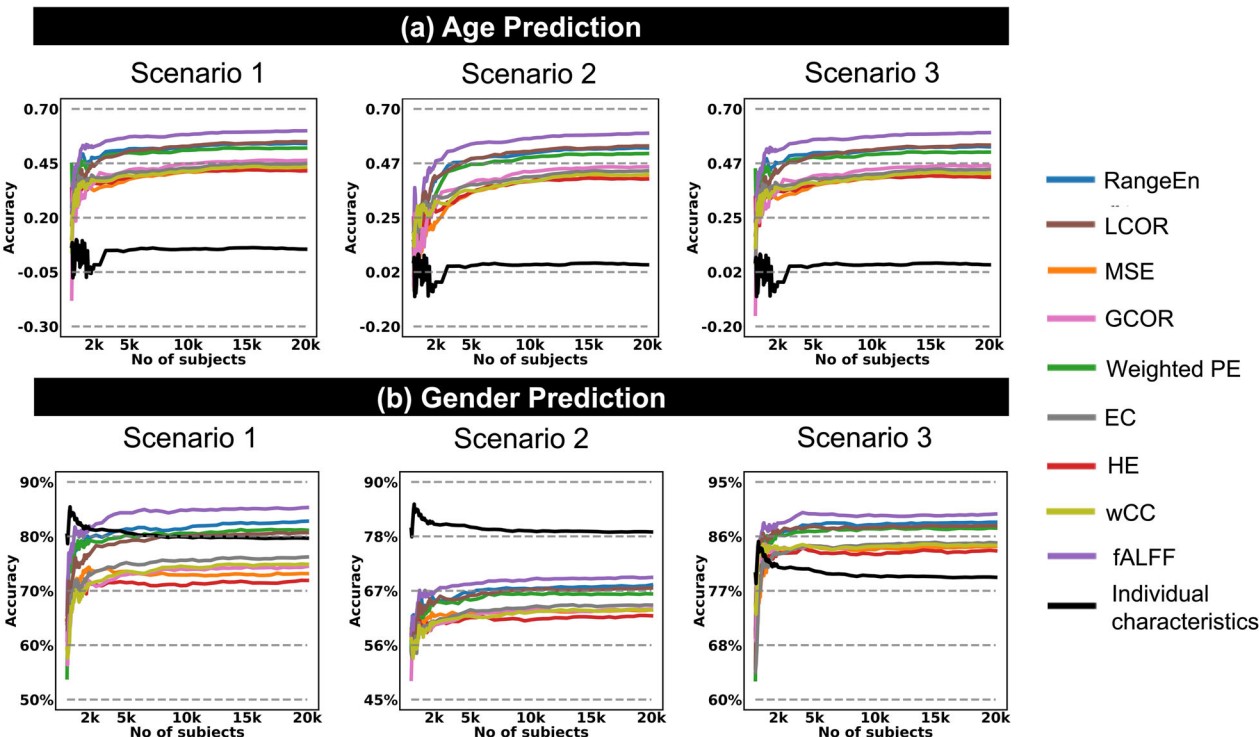

**Fig. 4 | Prediction of age and gender.** Prediction accuracy scores are associated with nine rsfMRI features and age and gender as targets using scenarios 1–3 of this study (panels (**a**) to (**c**), respectively) using the Schaefer400 brain atlas and ridge regression predictive modeling (see also Fig. 1b.1–b.3) and Methods). For age prediction, we considered gender and TIV as confounds, while for gender prediction, we considered age and TIV as confounds. Age prediction accuracies were computed as the Spearman correlation between the actual values and predicted values through SVM modeling. Gender prediction accuracies were computed as the balanced accuracy through SVM binary classification. Each rsfMRI feature is illustrated in a distinct color and listed in the figure legend. The population sizes from 100 to 2000 were increased with a 50-step increment and from 2000 to 20,000 with a 500-step increment. Abbreviations: rsfMRI resting state functional magnetic resonance imaging, TIV total intracranial volume, PE permutation entropy, fALFF fractional amplitude of low frequency fluctuations, LCOR local correlation, GCOR global correlation, EC eigenvector centrality, CC clustering coefficient.

different. For instance, *fALFF* and *wPE* were frequently among the most predictive features across all analysis scenarios. Therefore, we quantified the similarity between rsfMRI features using an individual identification paradigm (see Methods). A number of rsfMRI feature pairs showed a high level of match across subjects (Fig. 5). The pairs *wCC-EC*, *wPE-RangeEn$_B$*, *fALFF-LCOR*, and *MSE-HE* were among the most highly matched. The identification accuracy increased when age, gender, and TIV were either removed or when rsfMRI features were combined with individual characteristics (Fig. 5b–d). Importantly, identification accuracy decreased as the number of subjects increased, as opposed to the increase in prediction accuracy (Fig. 2).

## Discussion

A primary goal of neuroscience is to investigate the relationship between brain dynamics and individual differences in behavior[43]. Spontaneous fluctuations in blood oxygenation level-dependent (BOLD) changes measured by fMRI have been shown to exhibit complex and balanced dynamics in the time domain referred to as temporal complexity[44,45]. The interactions between BOLD changes across brain areas, also known as functional connectivity, provide useful perspectives on brain activity at a large scale. It has been demonstrated that these functional interactions are crucial for accomplishing mental tasks and are related to behavioral phenotypes[46].

Predictive modeling of neuroimaging data can provide individualized insights that greatly benefit personalized medicine[47,48]. This approach is valuable because it considers the natural variations in human cognition and brain function. Interventions and treatments customized to a person's

unique cognitive strengths and weaknesses can be informed by modeling of the relationship between individual brain dynamics and behavior. We can move away from one-size-fits-all methods and provide more accurate assessments and interventions by creating models that take this diversity into account.

The goal of the present study was to compare the capacity of rsfMRI features with age, gender, and TIV for predicting behavioral measures. To this end, we looked at how well four behavioral phenotypes, fluid intelligence, processing speed, and visual/numeric memory characteristics, can be predicted using various aspects of rsfMRI dynamics measured by temporal complexity/functional connectivity features. We demonstrated that, despite having different mathematical definitions, temporal complexity and functional connectivity features lead to comparable performance across a wide range of sample sizes[20]. The results were robust over all combinations of two brain parcellation atlases (Schaefer400 and Glasser360) and two predictive models (ridge regression and linear SVM).

Comparing MRI modalities for behavioral prediction has been the subject of several recent studies[49–51]. However, many of these studies have utilized the same datasets, primarily the widely used Human Connectome Project (HCP) database[52], with a medium sample size of fewer than 1500 participants. Behavioral prediction studies that use big datasets such as the UK Biobank database are still limited in the literature[14,40,42]. This makes extrapolating the findings of small sample size studies to larger sample sizes challenging[7]. Studies on reproducible brain-wide associations have also been established to require the involvement of thousands of participants[15]. To take these issues into account, we used a sizable portion of the UK Biobank

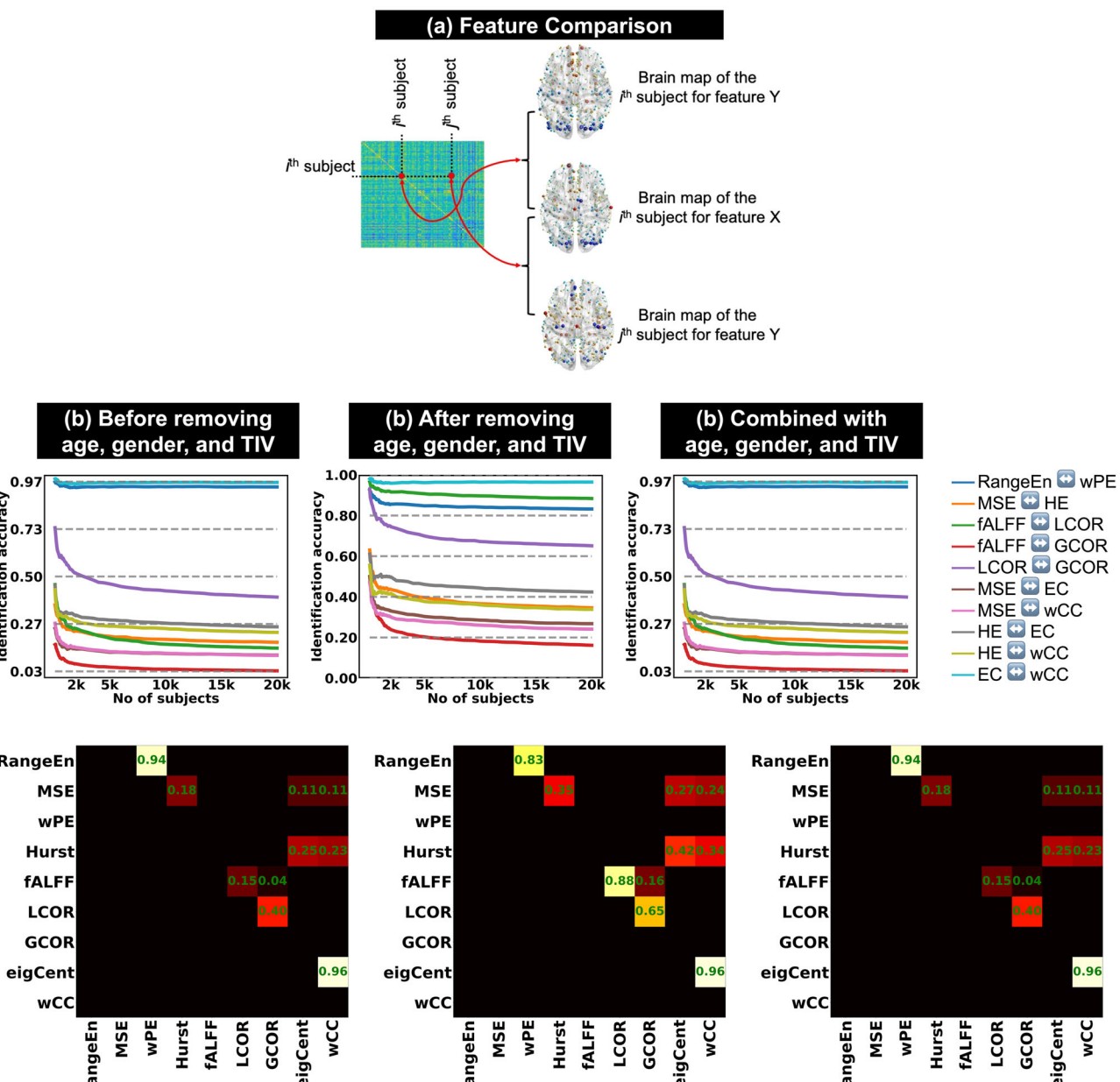

**Fig. 5 | Feature matching of rsfMRI.** The extracted features were obtained using the Schaefer400 brain atlas and fed into linear SVM predictive modeling. **a** A schematic example of comparing two rsfMRI features X and Y from the same subject in a sample. This comparison leads to the computation of an identification accuracy score (see Methods). **b–d** Identification accuracy patterns of 10 rsfMRI feature pairs with above zero matching are associated with three analysis scenarios of this study (see Fig. 1 as well as Methods). Each pair in the middle row panels has been depicted in a distinct color, and all pairs are listed in the figure legend. In each figure panel, the x-axis represents the population size in the analysis, and the y-axis shows the identification accuracy. The identification analyses were repeated for different sample sizes in the UK Biobank, ranging from $N_{subject} = 100$ to $N_{subject} = 20,000$. The population sizes from 100 to 2000 were increased with a 50-step increment (see the light orange shadow in the figure panels) and from 2000 to 20,000 with a 500-step increment (see the light blue shadow in the figure panels). The color-coded matrices in the row illustrate the identification accuracy of rsfMRI feature pairs for $N_{subject} = 20,000$. Abbreviations: rsfMRI resting state functional magnetic resonance imaging, ROI region of interest, tSNR temporal signal to noise ratio, TIV total intracranial volume, wPE weighted permutation entropy, MSE multiscale entropy, RangeEn range entropy, fALFF fractional amplitude of low frequency fluctuations, LCOR local correlation, GCOR global correlation, eigCent/EC eigenvector centrality, wCC weighted clustering coefficient.

database[20], varied the sample size from 100 to 20,000 subjects, and examined the effects of sample size scaling on the predictive capacity of rsfMRI temporal complexity and functional connectivity-derived measures for behavioral phenotypic prediction. Increasing the sample size improved the predictive accuracy, though it reached to a plateau after a certain sample size (roughly, at about $N_{subject} > 2000$). This is in line with the previous cognition

prediction studies using the UK Biobank and other rsfMRI databases showing that the accuracy of ridge regression modeling reaches a plateau with increasing sample size[40,53]. It emphasizes the need to consider the possibility of variation in the prediction accuracy of behavioral measures when studying small populations. The predictive ability of rsfMRI temporal complexity and functional connectivity features stems from the

interconnected neural activity, particularly the spontaneous BOLD fluctuations across brain regions[46]. Notably, *fALFF*, *wPE*, and AUC of *RangeEn*$_B$ consistently outperformed others in predicting behavioral measures and demographics. Our identification analysis emphasized the nuanced relationship between diverse features, revealing that certain features, like *fALFF* and *wPE*, consistently exhibit robust correlation at the individual level (Fig. 5). Noteworthy feature pairs, such as *wCC-EC* and *wPE-RangeEn*$_B$, showed high similarity but retained distinct information. Altogether, it suggests that although certain rsfMRI features may have some similarities across individuals, they also possess distinct information as demonstrated by their varying predictive abilities. Such a relationship has also been seen in the characteristics of neurological conditions such as epilepsy, which has been described as both a disorder of functional networks in the brain and an abnormality of its dynamics at the same time[54].

Combining functional connectivity and temporal complexity measures may provide improved prediction of behavioral phenotypes. However, it is essential to consider potentially high similarity between some of these rsfMRI measures as shown by our identification analysis. Highly correlated features can introduce multicollinearity which may pose challenges for prediction models, in turn making it challenging to discern contribution of individual features and resulting in less interpretable models. To effectively leverage the benefits of combining functional connectivity and temporal complexity measures, one may consider employing proper techniques that can deal with the multicollinearity. While combining these measures could indeed have implications for predictive modeling, we believe that investigating the best strategies for combining these measures, addressing potential multicollinearity, and optimizing predictive models extends beyond the scope of our current study. A comprehensive exploration of such strategies would require dedicated investigation and represents a valuable avenue for future research.

The combination of functional connectivity and temporal complexity measures with age, gender, and TIV could synergistically enhance predictive capacity, with individual characteristics providing additional information that complements rsfMRI features towards a more robust modeling. In contrast, removing individual characteristics using linear regression may simplify the modeling task by revealing independent individual signals, eliminating potential interactions, and finally, improving identification accuracy. It can also let the predictive model focus on the specific information that functional connectivity and temporal complexity have collected, leading to more precise identifications.

Depending on the analysis workflows of this study, we either used rsfMRI features, individual characteristics, or both as input features for behavioral prediction. The prediction accuracy using age, gender, and TIV was higher than that of all rsfMRI features. This finding highlights the importance of considering individual characteristics in the prediction of behavioral measures, especially when the goal is to maximize predictive accuracy. Additionally, this finding aligns with a prior study in which the integration of MRI data and sociodemographic factors systematically improved the accuracy of predicting age, fluid intelligence, and neuroticism[19]. It is also consistent with previous research in the ADHD-200 Global Competition, which found that when performing classification of ADHD diagnostics, individual characteristic data (site of data collection, age, gender, handedness, performance IQ, verbal IQ, and full scale IQ) performed better than a variety of fMRI features[18]. Overall, the three individual characteristics performed better than the combined feature vectors of size 403 and the 400 rsfMRI features. It implies that neuroimaging-based features offer additional information to the demographic and sociodemographic factors for predicting behavior. The combination of individual characteristics and rsfMRI features may offer an opportunity to harness the complementary information provided by both types of data. Further research is needed to explore the potential benefits of integrating individual characteristics with rsfMRI features in predictive models for behavioral phenotypes. Individual characteristics may also be handled as potential confounds when investigating associations between brain activity and behavior. Additional research into the association between rsfMRI features and individual characteristics in predictive models is needed to see if this relationship is vulnerable to confound removal.

Previous studies have shown that some behavioral phenotypes can be predicted better than others using neuroimaging data[7]. This is supported by our prediction results which show that regardless of the rsfMRI features used and the sample size, the processing speed measure was usually predicted better than the visual and numerical memory scores. A recent review of human fluid intelligence prediction using neuroimaging data has reported an average correlation of 0.15 with a $CI_{95\%}$ of [0.13, 0.17] across the fMRI literature[55]. This is confirmed by our fluid intelligence prediction results with a maximum correlation score of 0.23 using combined *LCOR* and individual characteristics and at very high sample sizes. Contrary to behavioral phenotypes, age and gender were easier to predict using both temporal complexity and functional connectivity features, as shown by a comparison between the prediction accuracy curves. However, gender prediction using age and TIV was better than age prediction using gender and TIV. All four analysis workflows passed a sanity check using the chance-level prediction of yesterday's fish consumption.

The issue of noise and artifacts can influence fMRI features, for example in *fALFF* which utilizes bandpass filtering of fMRI time series[31]. Temporal SNR is a metric for comparing the strength of an interest signal to the amount of background noise in the time domain[56]. The tSNR analysis results of our study indicate that even the rsfMRI features of brain regions with a relatively low tSNR, which are typically found in deeper areas of the brain and close to sinus cavities, still contain predictive information about cognition. It is corroborated by our observation that using more brain regions, even when their tSNR is rather low, leads to higher accuracy. To retain the same number of brain regions across individuals, we used the group-mean tSNR map of the full sample with 20,000 UK Biobank individuals to threshold subject-specific rsfMRI feature brain maps. This is because tSNR brain maps do not always agree on the same brain regions across participants. We believe that the information in this group-mean tSNR map from a very large sample is so compressed and dimensionally reduced that any influence of data leakage would be minimal.

Our results show a relatively inverse association between identification accuracy and prediction accuracy across different sample sizes. While adding more subjects improved behavioral phenotypic prediction accuracy, doing so reduced identification accuracy. This shows that the identification problem becomes harder as the sample size grows because there are more chances of obtaining a match with another subject than with the self. On the other hand, the prediction problem becomes relatively easier for larger populations because more information is available for learning. Our results also suggest that the nine investigated rsfMRI features can be categorized into various matched pairs. High identification accuracy between *wPE* and *RangeEn*$_B$, *HE* and *MSE*, *fALFF* and *LCOR*, and *wCC* and *EC*, are most notable. The similar prediction performance of these feature pairs can be partly explained by the measurements yielding similar individual-level patterns (as shown by the high identification accuracy), even though they are conceptually different.

There are a number of limitations that should be considered before arriving at a solid conclusion. First, we only used two linear predictive modeling algorithms in this study, so other models might capture different information. Second, even though many more variables, such as handedness and genetic factors, could influence the rsfMRI features, we only considered three individual characteristics in our predictive modeling. Third, we attempted to address the challenge of accurately quantifying behavioral phenotypes using some of the most reliable behavioral phenotypes available in the UK Biobank[41]. Despite that, these quantitative scores might still be unreliable and subject to oversimplification[57]. Unfortunately, standardized normed scores that account for demographic factors such as age and gender are not available in the UK Biobank database. Fourth, we only included cortical areas in our analyses. Using subcortical and cerebellar areas may provide a more complete picture.

Taken together, imaging-derived features such as rsfMRI temporal complexity and functional connectivity measures offer complementary

insights to demographic factors in predicting behavior and cognition. When constructing predictive models, it is recommended to thoughtfully choose and prioritize rsfMRI features based on their effectiveness in predicting the target behavioral phenotypes. Furthermore, the incorporation of demographic indicators such as age, gender, and TIV is advisable, particularly when striving for heightened prediction accuracy.

## Methods

### Data and preprocessing
We used the rsfMRI data of 20,000 unrelated UK Biobank participants after excluding subjects with mental and behavioral disorders (ICD10, category F), diseases of the nervous system (ICD10, category G), and cerebrovascular diseases (ICD10, categories I60 to 69). Data management of the UK Biobank datasets was performed using DataLad[58] on JURECA, a pre-exascale modular supercomputer operated by the Jülich Supercomputing Center at the Forschungszentrum Jülich, Germany. The duration of each rsfMRI scan was 6 minutes (490 time points), with a repetition time (*TR*) of 0.735 seconds, an echo time (*TE*) of 39 milliseconds, a voxel size of $2.4 \times 2.4 \times 2.4$ millimeters, and a field of view of $88 \times 88 \times 64$. The following procedure was performed on the rsfMRI datasets as part of a pipeline developed on behalf of the UK Biobank[17]: grand-mean intensity normalization of the entire 4D fMRI dataset by a single multiplicative factor; highpass temporal filtering using Gaussian-weighted least-squares straight line fitting with $\sigma = 50$ seconds; echo planar imaging unwarping; gradient distortion correction unwarping; and structured artifact removal through independent component analysis (ICA), followed by an ICA-based X-noiseifier (ICA-FIX)[59–61]. No spatial or temporal smoothing was applied to the fMRI volumes. The preprocessed data files, referred to as *filtered_func_data_clean.nii* in the UK Biobank database, were normalized to the MNI space using FSL's *applywarp* function with spline interpolation and parcellated using the Schaefer brain atlas into 400 ROIs (Schaefer400)[62] and using the Glasser brain atlas into 360 ROIs (Glasser360)[63]. Since we needed a continuous fMRI time series for the extraction of temporal complexity features, we did not apply motion scrubbing. Finally, we considered age, gender, and TIV as individual characteristics in the analyses and incorporated them into four analysis scenarios illustrated in Fig. 1. The TIV of each subject was extracted after brain extraction from the T1 image using the Computational Anatomy Toolbox (CAT12) for SPM[64].

Four behavioral phenotypes were selected as the predictive targets among the most reliable UK Biobank behavioral phenotypes, including fluid intelligence (data field 20016), processing speed (data field 20023), numeric memory (data field 20240), and visual memory (data field 399)[41]. Additionally, an unrelated binary target (fish consumption yesterday—data field 103140) was used as a sanity check of the rsfMRI features in the predictive modeling scenarios.

### Temporal complexity features
*HE*[35] is used to determine whether a time series contains a *long-memory process*. It quantifies three different types of trends: (*i*) values between 0.5 and 1, indicating that the time series is complex (balanced in time) and has long-range dependence; (*ii*) values less than 0.5, indicating that the time series is random and has short-range dependence; or (*iii*) a value close to 0.5, indicating that the time series is a random walk with no memory of the past. *HE* has been shown to be stable and reproducible across different fMRI datasets[65]. In this study, we estimated *HE* using the rescaled range analysis technique[35]. The *wPE*[36] is a modified version of permutation entropy[66], that captures order relations between time points in a signal and generates an ordinal pattern probability distribution using an embedding dimension *m* and a time delay $\tau$, where the former is the length of the patterns, and the latter is a lag parameter denoting the number of time points to shift throughout the time series. In this study, we used the parameters $m = 4$ and $\tau = 1$ and normalized the *wPE* values by dividing them by $log_2(m!)$ in order to get the numbers between 0 and 1. *RangeEn* offers two versions ($RangeEn_A$ and $RangeEn_B$) as modifications to approximative entropy[67] and sample entropy[68], respectively. A property of *RangeEn* is that regardless of the

nature of the signal dynamics, it always reaches 0 at its tolerance value of $r = 1$[37]. In light of this, one can obtain a complete trajectory of signal dynamics in the *r*-domain using this measure. Therefore, we extracted this trajectory from each ROI-wise rsfMRI time series and reduced its dimensionality by computing the area under its curve along the *r*-axis ($m = 2$). We have already shown that *RangeEn* is robust to variations in signal length[37], making it a viable option for relatively short-length time series such as rsfMRI. *MSE* is an extension of sample entropy that provides insights into the complexity of rsfMRI fluctuations over a range of time scales[38]. The measure returns a trajectory of sample entropy values across the time scales 1 to $\tau_{max}$. We have already shown that *MSE* of rsfMRI may be linked to higher-order cognition[27]. In this study, we chose the parameters $m = 2$, $r = 0.5$, and $\tau_{max} = 10$ for *MSE*, as a measure of temporal complexity. We then reduced its dimensionality by taking the area under its curve and dividing by $\tau_{max}$. This analysis was performed on parcellated brain regions using the Schaefer400 and Glasser360 brain atlases.

### Functional connectivity features
We computed the functional connectivity measures of rsfMRI at two spatial scales: (i) at the ROI level (*EC*, *wCC*) and (ii) first at the voxel level, then averaged within the ROIs (*fALFF*, *LCOR*, *GCOR*). For the ROI-wise measures, we characterized functional connectivity between every pair of ROIs in each rsfMRI dataset using a total of 400 regions for the Schaefer400 brain atlas[62] and 360 regions for the Glasser brain atlas (Glasser360)[63] and extracted the connections using Pearson correlation between mean fMRI time series[34]. *GCOR* serves as a representative of brain-wide correlation properties and a voxel-level representation of node centrality[33]. *LCOR* measures voxel-level local coherence defined as the average of the correlation coefficients between a voxel and its immediate surroundings (a Gaussian kernel with FWHM of 25 mm)[32]. Similar to *GCOR*, *LCOR* takes both the strength and sign of functional connections into consideration. *fALFF* quantifies the contribution of low frequency fluctuations to the total frequency range within a given frequency band (here, 0.008-0.09 Hz[31]). While *GCOR and LCOR* assess the strength of interregional and local cooperation by measuring the temporal similarity between voxels, *fALFF* evaluates the amplitude of regional neuronal activity. For each subject, the voxel-wise *GCOR*, *LCOR*, and *fALFF* brain maps were parcellated using both brain atlases[62,63]. *EC* is an ROI-based measure that indicates the impact of an ROI on the functional brain network[34]. The *EC* of the $i^{th}$ ROI corresponds to the $i^{th}$ element in the eigenvector corresponding to the largest eigenvalue of the ROI-wise functional connectome. *Weighted CC* quantifies how much the ROIs in the brain network functionally cluster together. This metric is calculated as the ratio of all triangles in which the $i^{th}$ ROI participates to all triangles that, theoretically, could be formed given the degree of the $i^{th}$ ROI's involvement in the brain's functional network[34]. The list of rsfMRI features in this study is summarized in Table S1 in the Supplementary Materials.

### tSNR analysis
In order to investigate the influence of tSNR levels of parcellated rsfMRI time series and the number of suprathreshold ROIs on the prediction accuracies of behavioral measures, we performed two analyses: one via tSNR thresholding and increasing the tSNR levels from 0% to 100% and another through decreasing the levels of tSNR from 100% to 0%. In fact, tSNR thresholding was used to control the number of suprathreshold ROIs, consequently influencing prediction accuracy. The rationale was that if the two prediction accuracies are similar, then the number of ROIs, and not the tSNR of the rsfMRI time series, is the primary determinant of the prediction performance. To this end, we used the Schaefer400 atlas and linear SVM with heuristic *C* across three scenarios for confound removal for all rsfMRI features and prediction targets at $N_{subject} = 20,000$. Finally, we plotted the prediction accuracies as a function of the number of suprathreshold ROIs as well as the tSNR levels. We calculated tSNR for each brain region as the ratio between the mean and standard deviation of its rsfMRI time series[56]. This led to a tSNR brain map for each participant, which we normalized over ROIs

and later averaged across the entire UK Biobank population. For both brain atlases, we used the group-average map for thresholding to exclude the noisiest brain regions at multiple tSNR levels from 0% threshold (i.e., preserving all ROIs for the prediction) to 100%, resulting in no suprathreshold ROIs. As an example, a tSNR thresholding level of 60% on the Schaefer400 brain atlas leads to 13 suprathreshold ROIs.

## Predictive modeling

Following previous studies in the field[7,40,42], we chose to use ridge regression with *L2*-norm regularization and classification for predictive modeling in this study. We also explored the robustness of our results by employing linear SVM with heuristic choice of the hyperparameter *C*. Linear SVM was chosen as it is a widely used machine learning technique for predictive modeling and can provide an alternative perspective on the relationships between demographic and anatomical factors and behavioral phenotypes. As illustrated in Fig. 1(b.1–b.4), we designed four analysis scenarios based on the role of individual characteristics in predictive modeling. We trained 78 ridge regression and linear SVM models for each behavioral phenotype on a wide range of UK Biobank subjects from $N_{subject} = 100$ to $N_{subject} = 2,000$ with a 50-step increment and from $N_{subject} = 2000$ to $N_{subject} = 20,000$ with a 500-step increment and each tSNR level, resulting in a total number of 36504 models (9 features × 4 targets × 78 population sizes × 13 tSNR levels) for each model type. We also trained 78 ridge binary classifiers for each model type using each rsfMRI feature to predict fish consumption yesterday (total number of models: 2×9126). In all cases, we estimated the best model hyperparameter $\lambda$ of the ridge regression/classification over the following values: [0, 0.00001, 0.0001, 0.001, 0.004, 0.007, 0.01, 0.04, 0.07, 0.1, 0.4, 0.7, 1, 1.5, 2, 2.5, 3, 3.5, 4, 5, 10, 15, 20, 30, 40, 50, 60, 70, 80, 100, 150, 200, 300, 500, 700, 1000, 10,000, 100,000, 1,000,000] through grid search. The hyperparameter *C* in linear SVM was calculated as $C = 1/\frac{1}{N}\Sigma_{n=1}^{N}\sqrt{\sum_{i=1}^{F}x_i^2}$ where *N* is the number of subjects, *F* is the number of features, and $x_i$ represents the *i* th z-scored feature value[69]. For the evaluation of prediction accuracy, we performed five repeats of 5-fold nested cross-validation using the scikit-learn[70] and Julearn (https://juaml.github.io/julearn/main/index.html) libraries in Python. For evaluation of the predictive models, we computed Spearman correlation coefficient between the actual targets and the model's predictions. To evaluate the binary classifications, we used balanced accuracy, a classification performance metric used to evaluate a model's predictive accuracy, particularly in situations where there is an imbalance in the distribution of classes, such as comparing non equisized populations of males and females. We repeated the predictive modeling for five targets (four behavioral phenotypes as well as fish consumption yesterday) and nine rsfMRI features at a range of sample sizes varying from 100 to 20,000. At each sample size, we randomly sampled the data to contain an equal number of males and females. We developed four predictive modeling scenarios based on the role of age, gender, and TIV in our study, as illustrated in Fig. 1. These scenarios included (*b.1*) prediction using rsfMRI features before removing individual characteristics, (*b.2*) prediction using rsfMRI features after treating individual characteristics as confounds and removing them, (*b.3*) prediction using combined rsfMRI features and individual characteristics, and (*b.4*) prediction using individual characteristics only. Individual characteristics were regressed out at the target level for regression modeling[5,7] and at the feature level for the classification analyses[16,71] using linear regression. Confound removal was performed in a cross-validation consistent manner to avoid data leakage[16].

## Feature comparison via identification analysis

We adapted the individual *identification paradigm* from the functional connectome fingerprinting literature[9,72] and applied it to comparing different rsfMRI features of the same subject across a population. In this context, identification refers to the process of identifying a rsfMRI feature vector (brain map) X having the highest spatial correlation with Y, one of the other eight rsfMRI feature maps across the entire population. The identification accuracy was defined as the proportion of correctly identified

individuals based on matching their two rsfMRI features. The score ranges between 0 and 1, with higher values indicating a better match. Individual characteristics were removed from the rsfMRI features at the ROI-level using linear regression (Fig. 5a).

## Data availability

UK Biobank data can be obtained via its standardized data access procedure (https://www.ukbiobank.ac.uk/). The subject IDs of the UK Biobank database whose data have been used in this study can be obtained from the first author (A.O.) upon request. Per the policies of the UK Biobank, the findings of this study, which utilize imaging and behavioral data, must be submitted to the UK Biobank team and can be acquired directly from them. All the analysis results have also been archived on the servers of Research Center Jülich.

## Code availability

Necessary Python codes for reproducing the analysis results of this study using the UK Biobank datasets are available at: https://github.com/omidvarnia/Dynamic_brain_connectivity_analysis.

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

## Acknowledgements

This research has been conducted using the UK Biobank Resource under Application Number 41655. The work was supported by the Schwerpunktprogramm (SPP2041), project number 454012190, EI 816 28-1 (Machine-learning on Brain Connectomics: Individual Prediction of Cognitive Functioning in Health and Cerebral Small Vessel Disease), and the Helmholtz Portfolio Theme "Supercomputing and Modeling for the Human Brain". SBE acknowledges funding by the European Union's Horizon 2020 Research and Innovation Program (grant agreements 945539 (HBP SGA3) and 826421 (VBC)) and the Deutsche Forschungsgemeinschaft (DFG, SFB 1451 & IRTG 2150). All data analyses for this study were performed on the JURECA pre-exascale modular supercomputer operated by the Jülich supercomputing center at Forschungszentrum Jülich, Germany.

## Author contributions

Each author made a substantial contribution to this study. The final draft of the manuscript was read and approved by all authors. We use the CRediT contributor role taxonomy to describe individual contributions to the paper. Conceptualization: A.O., L.S., D.I.L., S.B.E., K.R.P.; Data Curation: A.O., L.S., F.H., J.K., J.D.; Formal analysis: A.O.; Funding acquisition: S.B.E., K.R.P., G.T., B.C.; Methodology: A.O., L.S., S.B.E, K.R.P; Software: A.O., L.S., F.R., K.R.P.; Supervision: S.B.E., K.R.P.; Visualization: A.O.; Writing—original draft: A.O.; Writing—review & editing: A.O., L.S., D.I.L., F.R., F.H., J.K., J.D., M.P., G.T., B.C, S.B.E., K.R.P.

## Funding

## Competing interests

G.T. has received fees as consultant or lecturer from Acandis, Alexion, Amarin, Bayer, Boehringer Ingelheim, BristolMyersSquibb/Pfizer, Daichi Sankyo, Portola, and Stryker outside the submitted work. All other authors declare no competing interests.
