## [Peer Review File · Communications Biology]

Reviewers' comments:

Reviewer #1 (Remarks to the Author):

In this work, Omidvarnia and colleagues use a large-scale fMRI dataset of 20,000 individuals (UK Biobank) to investigate the performance of models based on various measures of functional connectivity (FC) and temporal complexity (TC) predicting four different cognitive phenotypes and the effects of individual characteristics, such as age, gender and total intracranial volume (TIV) on the performance.

The authors did a good job using a large dataset, testing many rsfMRI measures in various subsample sizes and at various number of regions based on tSNR thresholding which enabled them to compare an impressive number of models. There are, however, a few points the authors may wish to address and clarify.

My comments:

1. I would suggest that the authors are more consistent and focused on their main goals in the study. For example, in the Discussion section, the authors emphasise that their goal was to assess the capacity of FC and TC features to predict cognitive phenotypes but the title of the study and the abstract (in which they first state “It is unclear, however, to what extent rsfMRI carries independent information from the individual characteristics that is able to predict cognitive phenotypes.”) are focused more on the effects of individual characteristics on the predictions. The authors may want to keep the direction of their study clear - e.g., state all goals, but focus on the primary one stated in the title and abstract.
2. It would be helpful if the authors discuss what the implications of the study are in more depth. For example, what are their recommendations for prediction of cognitive phenotypes using rsfMRI? Which features are the best? Is there any point of using the brain measures when individual characteristics perform better? Should the individual characteristics be included in the models?
3. The Results and Methods sections both mention that Spearman’s correlation coefficient was used to assess the accuracy but the captions of Fig. 2 and Fig. 3 mention Pearson’s correlation coefficient. Could the authors clarify this? If Spearman’s coefficient was used, it would be helpful to explain on which values it was used (e.g., whether the order of individuals was used instead of the actual scores).
4. It would be informative to include some more details on the parcellation in the results, especially for the first set of results. For example, a reader may want to know that the initial results are based on all 400 regions which are later thresholded.
5. The caption of Fig. 2 mentions light orange and light blue shadows but those do not seem to be in the figure. The caption also mentions that the prediction accuracy improved with the number of suprathresholded ROIs but Fig. 2 does not seem to include such information. The authors may want to correct that.

6. For the analysis in Fig. 3, the text in the Results section says that prediction accuracy improves with the number of suprathresholded ROIs but the subtitle is “tSNR plays no major role”. The authors may want to unify and clarify these statements. If the point is that the difference in prediction is caused by the number of regions only (not by tSNR), it would be helpful to show that (by, for example, using random regions for predictions, not only based on the level of tSNR).
7. Related to 6., if the point is that the prediction improves with better tSNR, could it be somewhat quantified to what extent the prediction accuracy improved (in terms of the number of regions or tSNR)? In Fig. 3 the improvement does not seem much gradual after a certain point (no. of ROIs) so the authors may want to look into that.
8. I would also suggest that for better comparison and assessment of whether tSNR plays a role, the authors may also want to compare the performance of models based on the same number of regions but with different levels of tSNR (in case there is enough variance of tSNR across the regions).
9. Regarding the tSNR, it would be helpful to provide a visualisation - a brain map - indicating the tSNR levels in regions or voxels.
10. It is also unclear to me what the threshold of 65 % means and how that resulted in 14 ROIs. Is that 65 % of the maximum tSNR? Also, as in 9., it would be helpful to visualise those regions (after the maximum thresholding) as well.
11. The analysis of individual identification using different pairs of features is interesting. Could the authors add more on implications of the analysis in Fig. 5? For example, could we use a combination of the FC and TC measures for a better prediction? If so, given that some of them seem to be highly correlated, is there any point of using them together?
12. Along with 11., in the last sentence in the Discussion, the authors mention that the study could help create better individualised models - it would be helpful to comment more on why individualised models in particular.
13. In Fig. 5, it is interesting that individual identification accuracy in which FC and TC measures are combined with age, gender and TIV or in which those characteristics are removed is better than when the characteristics are not removed? Could the authors provide some explanation on how that is possible?
14. Finally, I would suggest the authors make the abstract a little clearer and with the main goals clearly stated (related to 1.). For example, they may want to modify and clarify the subject in the sentence ‘It is also consistent across different levels...’.

Reviewer #2 (Remarks to the Author):

General comments

Are the results dependent on the atlas/regions of interest chosen or are they generalizable beyond the atlas chosen?

Are the four cognitive phenotypes general enough to represent cognition? Related to this, the authors choose these targets as they are “the four most reliable cognitive phenotypes”. What are the results for unreliable phenotypes? Could they provide a set of results for one unreliable phenotype?

What was the motivation for using ridge regression (apart from it being used by other researchers)? It is well known in ML/statistics literature that there exists superior methods to ridge regression. Related to my first comment, are the results dependent on the method chosen or are they generalizable beyond ridge regression?

On P.4, the authors state that they used the Spearman correlation between the real and predicted targeted but in the caption in Figure 2 they state Pearson correlation.

On P.6 (last line), they authors state Figure 1 – D.1. Where is Figure D.1? D is also mentioned in the first paragraph of P.7.

In the last paragraph of P.7, the authors state that they show “prediction accuracies ... to 60%” in Figure 2. This is not evident. There is also inconsistency in the percentage in the caption of Figure 3 and the figure.

The results in Figure 3 should be explained in more detail in the text.

Why are the results from predicting TIC not included in Figure 4?

What is the definition of balanced accuracy?

In the Discussion, the authors consider the problem of dataset decay. But this would only be applicable to inference or testing, which is not the focus of this paper (prediction).

The grid search for the hyperparameter λ should be increased to cover more values.

Response to Reviewers

The authors wish to thank the co-editor-in-chief and the reviewers for their insightful and helpful comments. A modified version of the manuscript is submitted as revised in accordance with the reviewers' feedback which incorporates a substantial number of additional analyses. In this document, we address each of the reviewers' comments separately and in detail. Remarks and questions are presented in bold and our response to each comment along with the action taken is in red font directly below them. In addition, we have marked the updated parts of the revised document in red to facilitate easier tracking.

Reviewer 1

Comment 1: I would suggest that the authors are more consistent and focused on their main goals in the study. For example, in the Discussion section, the authors emphasise that their goal was to assess the capacity of FC and TC features to predict cognitive phenotypes but the title of the study and the abstract (in which they first state “It is unclear, however, to what extent rsfMRI carries independent information from the individual characteristics that is able to predict cognitive phenotypes.”) are focused more on the effects of individual characteristics on the predictions. The authors may want to keep the direction of their study clear - e.g., state all goals, but focus on the primary one stated in the title and abstract.

Response and action: We appreciate the reviewer's useful feedback and have taken the following actions to improve the consistency and clarity of the manuscript. Our primary goal in this study was to compare imaging-derived brain function features with individual characteristics (ICs) to predict behavioural measures. While it is true that every feature can make a prediction to some extent, the main question was whether resting-state functional MRI or rsfMRI adds value. We have specifically revised the Title as well as the Abstract and Discussion sections to ensure that the primary goal, as stated in the title and abstract, emphasizing the crucial role of individual characteristics in predicting behavioural phenotypes, remains the central theme of the study. This adjustment is intended to offer readers a clearer and more focused narrative. The modifications include:

- **Modified Title:** “Comparison between resting-state fMRI features and individual characteristics for the prediction of behavioural phenotypes: a machine learning perspective”
- **Modified Abstract:** “In this research, we aimed to compare imaging-derived features of brain function, measured by resting-state fMRI (rsfMRI), with individual characteristics (ICs) such as age, gender, and total intracranial volume (TIV) to predict behavioral measures. Using a dataset of 20,000 healthy individuals from the UK Biobank, we developed a machine learning framework based on rsfMRI features, focusing on temporal complexity (TC) and functional connectivity (FC) measures. Our analysis across four behavioral phenotypes revealed that both TC and FC methods provide comparable predictive performance. However, ICs consistently outperformed rsfMRI features in predictive accuracy, particularly in analyses involving smaller sample sizes. Interestingly, integrating rsfMRI features with demographic data sometimes enhanced predictive outcomes. The efficacy of different predictive modeling techniques and the choice of brain atlas were also examined, showing no significant influence on the results. To summarize, while ICs are superior to rsfMRI in predicting behavioral phenotypes, rsfMRI still conveys additional predictive value in the context of machine learning, such as investigating the roles of specific brain regions or similar objectives.”
- **Modified Text (in the Introduction section):** “We found that the TC and FC features both give about the same predictive power when applied to four different behavioral phenotypes. Nevertheless, ICs routinely outperformed rsfMRI features in predictive accuracy in analyses involving smaller sample sizes. Age, gender, and TIV, when combined with rsfMRI features, improved predictive outcomes. The results were relatively unaffected by the selection of brain atlas or the effectiveness of two predictive modeling method. The removal of age, gender, and TIV from the features or targets resulted in reduced performance. The results also show that age and gender could be predicted more accurately than behavioral phenotypes in general. Our findings indicate that ICs outperform rsfMRI in predicting behavioral phenotypes, but rsfMRI can still offer supplementary predictive capability. Consequently, it is imperative for future studies to meticulously integrate these factors to enhance precision”.
- **Modified Text (in the Discussion section):** "The goal of the present study was to compare the capacity of rsfMRI features with age, gender, and TIV for predicting behavioral measures. To this end, we looked at how well four

behavioral phenotypes, fluid intelligence, processing speed, and visual/numeric memory characteristics, can be predicted by various aspects of rsfMRI dynamics measured by TC/FC features. We demonstrated that, despite having different mathematical definitions, the TC and FC features lead to comparable performance across a wide range of sample sizes [Miller et al., 2016]. The results were robust over all combinations of two brain parcellation atlases (Schaefer400 and Glasser360) and two predictive models (ridge regression and linear SVM).”

Comment 2: It would be helpful if the authors discuss what the implications of the study are in more depth. For example, what are their recommendations for prediction of cognitive phenotypes using rsfMRI? Which features are the best? Is there any point of using the brain measures when individual characteristics perform better? Should the individual characteristics be included in the models?

Response and action: Thank you for this useful suggestion. The key practical implications of our study are as follows:

- 1) **Comparable predictive performance:** TC and FC measures derived from rsfMRI show comparable predictive performance for behavioral measures. However, fALFF, wPE, and RangeEnB outperform other features.
- 2) **Demographic factors outperform rsfMRI:** ICs such as age, gender, and TIV consistently outperform rsfMRI features in predictive accuracy, especially with smaller sample sizes.
- 3) **Balancing ICs and rsfMRI for improved accuracy:** Integration of rsfMRI features with ICs enhances predictive outcomes, suggesting a complementary role for rsfMRI to the demographic factors for predicting behaviour.
- 4) **Consistent efficacy across techniques and atlases:** The choice of predictive modeling techniques and brain atlas does not significantly influence the results, indicating robustness across different approaches.

Considering this feedback, we have enhanced the Discussion section in the manuscript to address the following points:

- We have provided insights into which rsfMRI features performed best in predicting behavioural phenotypes and their potential utility in cognitive prediction tasks:

→ **Modified Text (in the Discussion section):** “Increasing the sample size improved the predictive accuracy, though it reached to a plateau after a certain sample size (roughly, at about $N_{\text{subjects}} > 2000$). This is in line with the previous cognition prediction studies using the UK Biobank and other rsfMRI databases showing that the accuracy of ridge regression modeling reaches a plateau with increasing sample size [Cui and Gong, 2018; He et al., 2020]. It emphasizes the need to consider the possibility of variation in the prediction accuracy of behavioral measures when studying small populations. The predictive ability of rsfMRI TC and FC features stems from the interconnected neural activity, particularly the spontaneous blood oxygenation fluctuations across brain regions [Liégeois et al., 2019]. Notably, fALFF, wPE, and AUC of RangeEnB consistently outperformed others in predicting behavioral measures and demographics. Our identification analysis emphasized the nuanced relationship between diverse features, revealing that certain features, like fALFF and wPE, consistently exhibit robust correlation at the individual level (Figure 5). Noteworthy feature pairs, such as wCC-EC and wPE-RangeEnB, showed high similarity but retained distinct information. Altogether, it suggests that although certain rsfMRI features may have some similarities across individuals, they also possess distinct information as demonstrated by their varying predictive abilities.”

- **The Role of Individual Characteristics:** We have discussed whether and when it is advantageous to include individual characteristics in predictive models alongside rsfMRI features, considering the scenarios in which these characteristics outperformed rsfMRI:

→ **Modified Text (in the Discussion section):** “Depending on the analysis workflows of this study, we either used rsfMRI features, ICs, or both as input features for behavioral prediction. The prediction accuracy using age, gender, and TIV was higher than that of all rsfMRI features. This finding highlights the importance of considering ICs in the prediction of behavioral measures, especially when the goal is to maximize predictive accuracy. Additionally, this finding aligns with a prior study in which the integration of MRI data and sociodemographic factors systematically improved the accuracy of predicting age, fluid intelligence, and neuroticism [Dadi et al., 2021]. It is also consistent with previous research in the ADHD-200 Global Competition, which found that when performing classification of ADHD diagnostics, individual characteristic data (site of data collection, age, gender, handedness, performance IQ, verbal IQ, and full scale IQ) performed better than a variety of fMRI features [Brown et al., 2012]. Overall, the three ICs performed better than the combined feature vectors of size 403 and the 400 rsfMRI feature vectors. It implies that imaging-based features offer additional information to the demographic and sociodemographic factors for

predicting behavior. The combination of ICs and rsfMRI features may offer an opportunity to harness the complementary information provided by both types of data. Further research is needed to explore the potential benefits of integrating ICs with rsfMRI features in predictive models for behavioral phenotypes. ICs may also be handled as potential confounds when investigating associations between brain activity and behavior. Additional research into the association between rsfMRI features and ICs in predictive models is needed to see if this relationship is vulnerable to confound removal.”

- We have elaborated on the practical implications of our findings for researchers and clinicians interested in using rsfMRI for cognitive prediction and emphasized the importance of considering individual characteristics in such applications:

→ **Modified Text (in the Discussion section):** "This finding emphasizes the need to consider the possibility of variation in the prediction accuracy of cognitive measures when using only rsfMRI features, as opposed to the more consistent performance seen when using ICs as input features."

→ **Modified Text (in the Discussion section):** “Taken together, imaging-derived features such as rsfMRI TC and FC measures offer complementary insights to demographic factors in predicting behavior and cognition. When constructing predictive models, it is recommended to thoughtfully choose and prioritize rsfMRI features based on their effectiveness in predicting the target behavioral phenotypes. Furthermore, the incorporation of demographic indicators such as age, gender, and TIV is advisable, particularly when striving for heightened prediction accuracy.”

Comment 3: The Results and Methods sections both mention that Spearman’s correlation coefficient was used to assess the accuracy but the captions of Fig. 2 and Fig. 3 mention Pearson’s correlation coefficient. Could the authors clarify this? If Spearman’s coefficient was used, it would be helpful to explain on which values it was used (e.g., whether the order of individuals was used instead of the actual scores).

Response and action: We apologize for the inconsistency in our reporting. In our study, we indeed used Spearman's correlation coefficient to quantify the accuracy of predictions, particularly in Figures 2 and 3. This choice was made to evaluate the monotonic relationship between predicted values and actual scores, considering the potential nonlinear associations that may exist. To clarify, Spearman's correlation coefficient was applied to assess the concordance between predicted and actual phenotype scores. This statistical is more robust to outliers and nonlinear associations in contrast to Pearson’s correlation coefficient. We have updated the captions of all figures to accurately reflect the use of Spearman's correlation coefficient.

Comment 4: It would be informative to include some more details on the parcellation in the results, especially for the first set of results. For example, a reader may want to know that the initial results are based on all 400 regions which are later thresholded.

Response and action: Thank you for the suggestion. We agree that providing more details about the parcellation approach can improve clarity. We would like to highlight a few points in response to your question: (1) In the first set of results, we indeed used all 400 regions of the Schaefer brain atlas. However, we added the Glasser atlas with 360 cortical regions to the new analyses in the revised manuscript. (2) We were also able to reproduce the previous set of Schaefer 400 atlas results using the Glasser 360 atlas for all prediction and identification accuracy analyses and using two modeling approaches: ridge regression and linear SVM. We have added additional information to the relevant section in the Methods and Results sections of the revised manuscript to clarify this aspect of our methodology. Here is the modified text:

→ **Modified Text (Methods Section):** "We computed the FC measures of rsfMRI at two spatial scales: (i) at the ROI level (EC, wCC) and (ii) first at the voxel level, then averaged within the ROIs (fALFF, LCOR, GCOR). For the ROI-wise measures, we characterized the FCs between every pair of ROIs in each rsfMRI dataset using a total of 400 regions for the Schaefer brain atlas [Schaefer et al., 2018] and 360 regions for the Glasser brain atlas [Glasser et al., 2016] and extracted the connections using Pearson correlation between mean fMRI time series."

→ **Modified Text (Methods Section):** "In this study, we chose the parameters $m = 2$, $r = 0.5$, and $\tau_{max} = 10$ for MSE, a measure of temporal complexity. We then reduced its dimensionality by taking the area under their curves and

dividing by t_{max} . It is worth noting that this analysis was performed on parcellated brain regions using the Schaefer and Glasser brain atlases."

- **Modified Text (Methods Section):** "For each subject, the voxelwise GCOR, LCOR, and fALFF brain maps were parcellated using both brain atlases [Glasser et al., 2016; Schaefer et al., 2018]."
- **Added Text (Results Section):** "The findings presented in Figure 2, along with Figures S1-S3, utilized all brain regions without applying any tSNR thresholding. This included a total of 400 and 360 regions for the Schaefer and Glasser atlases, respectively. Then, we only used the Schaefer brain atlas to investigate the impact of tSNR thresholding on the prediction accuracies (Figure 3 as well as Figures S18-S21). The brain feature maps were subsequently subjected to a thresholding process based on the ROI-wise tSNR values of parcellated rsfMRI datasets. The subset of ROIs that passed the threshold were retained as features for prediction analysis."

Comment 5: The caption of Fig. 2 mentions light orange and light blue shadows but those do not seem to be in the figure. The caption also mentions that the prediction accuracy improved with the number of suprathresholded ROIs but Fig. 2 does not seem to include such information. The authors may want to correct that.

Response and action: Thank you for the question. Please note that the reference to the light orange and light blue shadows in Figure 2 is accurate, since the Python codes have automatically generated the color legends for every figure. Furthermore, we acknowledge the missing information regarding the relationship between the number of suprathresholded ROIs and prediction accuracy in Figure 2 caption. The mention of relationship between the number of suprathresholded ROIs and prediction accuracy has now been removed from the corrected figure caption.

- **Modified caption for Figure 2:** "Prediction accuracy for behavioural phenotypes and ICs across different sample sizes. The x-axis represents the population size in the analysis, ranging from 100 to 20,000 participants. The y-axis shows the prediction accuracy measured by the Spearman correlation coefficient for behavioural phenotypes (fluid intelligence, processing speed, visual memory, and numeric memory) as well as age, and by the balanced accuracy for gender and fish consumption prediction. The prediction accuracy curves for each behavioural phenotype and individual characteristic are color-coded."

Comment 6: For the analysis in Fig. 3, the text in the Results section says that prediction accuracy improves with the number of suprathresholded ROIs but the subtitle is "tSNR plays no major role". The authors may want to unify and clarify these statements. If the point is that the difference in prediction is caused by the number of regions only (not by tSNR), it would be helpful to show that (by, for example, using random regions for predictions, not only based on the level of tSNR).

Related to 6., if the point is that the prediction improves with better tSNR, could it be somewhat quantified to what extent the prediction accuracy improved (in terms of the number of regions or tSNR)? In Fig. 3 the improvement does not seem much gradual after a certain point (no. of ROIs) so the authors may want to look into that.

I would also suggest that for better comparison and assessment of whether tSNR plays a role, the authors may also want to compare the performance of models based on the same number of regions but with different levels of tSNR (in case there is enough variance of tSNR across the regions).

Regarding the tSNR, it would be helpful to provide a visualisation - a brain map - indicating the tSNR levels in regions or voxels.

It is also unclear to me what the threshold of 65 % means and how that resulted in 14 ROIs. Is that 65 % of the maximum tSNR? Also, as in 9., it would be helpful to visualise those regions (after the maximum thresholding) as well.

Response and action: Thank you for the feedback. Here, we have put together all comments that we believe are related to the tSNR analysis part of our study and answer them all at the same time. To address these questions, we have done these extra steps:

- We conducted two analyses regarding the impact of the number of ROIs and tSNR thresholding on the prediction accuracies of all behavioural measures. The results confirmed that the accuracies are mainly driven by the number of ROIs and not, by the tSNR level of the rsfMRI time series (see Figures S16 and S17).

- We extracted the 2D color-coded maps of prediction accuracies versus tSNR thresholds for all features and all targets using both Schaefer and Glasser brain atlases as well as two modeling methods, i.e., ridge regression and linear SVM. The results have been illustrated in Figure 3 and Figures S18 to S21.
- We added the group level mean tSNR brain maps obtained from the Schaefer and Glasser brain atlases in **Figure S22**. According to this figure, the numbers of suprathreshold ROIs at tSNR threshold level spanning from 0% to 60% with 5% increment for the Schaefer and Glasser brain atlases are [400, 397, 397, 396, 387, 366, 333, 276, 201, 130, 68, 33, 13] and [360, 360, 356, 353, 338, 311, 270, 203, 142, 83, 57, 25, 10], respectively. For both brain atlases, tSNR levels above 60% led to no suprathreshold ROIs.
- We would like to correct our previous statement and make it more precise in a sense that the minimum number of suprathreshold ROIs in the Schaefer brain atlas occurs at the 60% tSNR level and is equal to N=13 (not 14). To have a complete picture, we increased the tSNR thresholding interval in the relevant figures (Figure 3 and Figures S16 to S21) from 0% (no threshold) to 100% (complete thresholding resulting in no ROIs).

The two analyses were as follows: one via tSNR thresholding and increasing the tSNR levels from 0% to 100% and another through decreasing the levels of tSNR from 100% to 0%. In fact, tSNR thresholding was used to control the number of suprathreshold ROIs, consequently influencing prediction accuracy. The rationale was that if the two prediction accuracies are similar, then the number of ROIs, and not the tSNR of the rsfMRI time series, is the primary determinant of the prediction performance. We chose to not conduct these analyses using randomized ROIs because incorporating randomization in the selection of ROIs could introduce complications and potentially complicate the interpretation of the specific effects we intended to clarify related to tSNR. We used the Schaefer 400 atlas and linear SVM with heuristic C across three scenarios for confound removal for all rsfMRI features and prediction targets at N=20,000. Finally, we plotted the prediction accuracies as a function of the number of suprathreshold ROIs as well as the tSNR levels. Figures S16 and S17 illustrate the relative independence of the prediction accuracies to the tSNR level. The corresponding panels of the two figures show a similar pattern of prediction performance, but in an inverse order. It suggests that the number of ROIs in the feature vectors is deriving the accuracy of the predictions, while the tSNR of the involved ROIs is not significant in this procedure. A more comprehensive view of the tSNR influence on the prediction accuracies can be observed in Figure 3 as well as Figures S18 to S21 where the varying impact of tSNR across the entire scaling range has been illustrated. The y-values in the 2D plots represent a certain number of ROIs and the change in the colours indicates how the prediction accuracies are influenced by the population size (horizontally) and the tSNR threshold level (vertically). We have explained two analyses and the rationale behind them in the Methods section and the findings in the Results section as follows:

- **Added Text (Methods Section):** "In order to investigate the influence of tSNR levels of parcellated rsfMRI time series and the number of suprathreshold ROIs on the prediction accuracies of behavioral measures, we performed two analyses: one via tSNR thresholding and increasing the tSNR levels from 0% to 100% and another through decreasing the levels of tSNR from 100% to 0%. In fact, tSNR thresholding was used to control the number of suprathreshold ROIs, consequently influencing prediction accuracy. The rationale was that if the two prediction accuracies are similar, then the number of ROIs, and not the tSNR of the rsfMRI time series, is the primary determinant of the prediction performance. To this end, we used the Schaefer400 atlas and linear SVM with heuristic C across three scenarios for confound removal for all rsfMRI features and prediction targets at N=20,000. Finally, we plotted the prediction accuracies as a function of the number of suprathreshold ROIs as well as the tSNR levels. We calculated tSNR for each brain region as the ratio between the mean and standard deviation of its rsfMRI time series [Murphy et al., 2007]. This led to a tSNR brain map for each participant, which we normalized over ROIs and later averaged across the entire UK Biobank population (N_{subject} = 20,000). For both brain atlases, we used the group-average map for thresholding to exclude the noisiest brain regions at multiple tSNR levels from 0% threshold (i.e., preserving all ROIs for the prediction) to 100%, resulting in no suprathreshold ROIs. As an example, a tSNR thresholding level of 60% on the Schaefer400 brain atlas leads to 13 suprathreshold ROIs."
- **Modified Text (Results Section):** "We then asked whether excluding brain regions with low tSNR levels would increase prediction accuracy. We used a group-level tSNR map to threshold the rsfMRI feature maps (see Methods). Figure 3 shows fluid intelligence prediction accuracies for scenarios 1 to 3 after stepwise thresholding on the tSNR maps from 0% (no threshold, equivalent with the results illustrated in Figure 2) to 100% (no ROI for prediction) with 5% increments. Each panel in the figure represents a distinct pair of features and targets, with color-coded accuracy values. The x-axis indicates the population size in the analysis, while the y-axis denotes the count of suprathreshold ROIs after tSNR thresholding. The predictive modeling for each feature-target pair was conducted across various sample sizes, spanning N_{subject} = 100 to N_{subject} = 20,000. For population sizes between 100 and 2000, increments of 50 subjects were employed, while increments of 500 subjects were applied for the range of 2000 to 20,000.

Prediction accuracies improved with increasing sample size and the number of suprathreshold ROIs. The finding was consistent across the other behavioral phenotypes (see Figure S18 to Figure S20). Prediction accuracy for fish consumption remained at chance-level for all tSNR thresholds (Figure S21). As shown in Figure 3, we observed that overall, prediction performance improved with the number of suprathreshold ROIs and the tSNR level. However, the figure does not explain which of these two factors are the main driver here. Figure S16 and Figure S17 show the prediction accuracies as a function of these two factors using the Schaefer400 brain atlas and the linear SVM model with heuristic C for all rsfMRI features and prediction targets at N=20,000. As Figure S16 illustrates, the prediction accuracies strongly depend on the number of ROIs when this number is low (typically less than 150). However, they increase and become largely independent of the number of ROIs when this number is sufficiently high. Given that the two sets of prediction accuracy curves (shown in Figure S16 and Figure S17) show a similar pattern of prediction performance, one can conclude that the ROI count is the main factor influencing the accuracy of the predictions, regardless of the tSNR of the involved ROIs in the prediction procedure. Applying a tSNR threshold of 60% resulted in 13 suprathreshold ROIs using the Schaefer400 brain atlas. This was the minimum number of suprathreshold ROIs that can be detected in the parcellated data following tSNR thresholding. No suprathreshold ROIs were produced when the ROI thresholds were less than 60% of the maximum tSNR across all regions. Figure S17 shows that most of the prediction accuracy curves drop sharply at the tSNR level of about 65%. This means that the feature vectors that have been thresholded above this level are not informative enough for prediction because there are not enough suprathreshold ROIs for that. However, the prediction accuracy curves have been presented in the figures from 0% to 100% of tSNR thresholding for the sake of completeness. Figure S22 illustrates the spatial distribution of the tSNR values across brain regions for both brain parcellation atlases. According to Figure S22, the numbers of suprathreshold ROIs at tSNR threshold level spanning from 0% to 60% with 5% increment for the Schaefer400 and Glasser360 brain atlases are [400, 397, 397, 396, 387, 366, 333, 276, 201, 130, 68, 33, 13] and [360, 360, 356, 353, 338, 311, 270, 203, 142, 83, 57, 25, 10], respectively. For both brain atlases, tSNR levels above 60% led to no suprathreshold ROIs. We have included the nifti files of normalized tSNR maps for both brain atlases in the supplement allowing the readers to directly examine any desired threshold."

- We have also updated the caption of **Figure 3** to align with the modifications made in the Results section:
- **Modified Caption for Figure 3:** "Spearman correlation accuracy associated with ridge regression modeling of fluid intelligence using nine rsfMRI features and after tSNR thresholding from 0% to 100%. The numbers of suprathreshold ROIs at tSNR threshold level spanning from 0% to 60% with 5% increment for the Schaefer and Glasser brain atlases are [400, 397, 397, 396, 387, 366, 333, 276, 201, 130, 68, 33, 13] and [360, 360, 356, 353, 338, 311, 270, 203, 142, 83, 57, 25, 10], respectively. For both brain atlases, tSNR levels above 60% led to no suprathreshold ROIs."
- We have explained how 60% threshold results in 13 ROIs in the Results section (see below). The Methods and Results sections have also been updated to reflect the above explanation as follows:
- **Modified Text (Methods section):** "For both brain atlases, we used the group-average map for thresholding to exclude the *noisiest* brain regions at multiple tSNR levels from 0% threshold (i.e., preserving all ROIs for the prediction) to 100%, resulting in no suprathreshold ROIs. As an example, a tSNR thresholding level of 60% on the Schaefer brain atlas leads to 13 suprathreshold ROIs."
- **Added Text (Results Section):** "Applying a tSNR threshold of 60% resulted in 13 suprathreshold ROIs in the Schaefer brain atlas. This was the minimum number of suprathreshold ROIs that can be detected in the Schaefer brain atlas following tSNR thresholding. No suprathreshold ROIs were produced when the ROI thresholds are less than 60% of the maximum tSNR across all regions. Figure S17 shows that most of the prediction accuracy curves drop sharply at the tSNR level of about 65%. This means that the feature vectors that have been thresholded above this level are not informative enough for prediction because there are not enough suprathreshold ROIs. However, the prediction accuracy curves have been presented in the figures from 0% to 100% of tSNR thresholding for the sake of completeness. Figure S22 illustrates the spatial distribution of the tSNR values across brain regions. We have included the nifti files of normalized tSNR maps for both brain atlases in the supplement allowing the readers to directly examine any desired threshold. According to this figure, the numbers of suprathreshold ROIs at tSNR threshold level spanning from 0% to 60% with 5% increment for the Schaefer and Glasser brain atlases are [400, 397, 397, 396, 387, 366, 333, 276, 201, 130, 68, 33, 13] and [360, 360, 356, 353, 338, 311, 270, 203, 142, 83, 57, 25, 10], respectively. For both brain atlases, tSNR levels above 60% led to no suprathreshold ROIs."

Comment 7: The analysis of individual identification using different pairs of features is interesting. Could the authors add more on implications of the analysis in Fig. 5? For example, could we use a combination of the FC and TC measures for a better prediction? If so, given that some of them seem to be highly correlated, is there any point of using them together?

Response and action: We appreciate the reviewer's positive feedback and interest. We have added this point in the Discussion:

→ **Added Text (Discussion Section):** "Combining FC and TC measures may provide improved prediction of behavioral phenotypes. However, it is essential to consider potentially high similarity between some of these rsfMRI measures as shown by our identification analysis. Highly correlated features can introduce multicollinearity which may pose challenges for prediction models, in turn making it challenging to discern contribution of individual features and resulting in less interpretable models. To effectively leverage the benefits of combining FC and TC measures, one may consider employing proper techniques that can deal with the multicollinearity. While combining these measures could indeed have implications for predictive modeling, we believe that investigating the best strategies for combining these measures, addressing potential multicollinearity, and optimizing predictive models extends beyond the scope of our current study. A comprehensive exploration of such strategies would require dedicated investigation and represents a valuable avenue for future research.

The combination of FC and TC measures with age, gender, and TIV could synergistically enhance predictive capacity, with ICs providing additional information that complements rsfMRI features towards a more robust modeling. In contrast, removing ICs using linear regression may simplify the modeling task by revealing independent individual signals, eliminating potential interactions, and finally, improving identification accuracy. It can also let the predictive model focus on the specific information that FC and TC have collected, leading to more precise identifications."

Comment 8: Along with 11., in the last sentence in the Discussion, the authors mention that the study could help create better individualised models - it would be helpful to comment more on why individualised models in particular.

Response and action: Thank you for this valid comment. We have expanded the last sentence in the Discussion as follows:

→ **Added Text (Discussion Section):** "Predictive modeling of neuroimaging data can provide individualized insights that greatly benefit personalized medicine [Michon et al., 2022; Tejavibulya et al., 2022]. This approach is valuable because it considers the natural variations in human cognition and brain function. Interventions and treatments customized to a person's unique cognitive strengths and weaknesses can be informed by modeling of the relationship between individual brain dynamics and behavior. We can move away from one-size-fits-all methods and provide more accurate assessments and interventions by creating models that take this diversity into account."

Comment 9: In Fig. 5, it is interesting that individual identification accuracy in which FC and TC measures are combined with age, gender and TIV or in which those characteristics are removed is better than when the characteristics are not removed? Could the authors provide some explanation on how that is possible?

Response and action: To address this comment, we updated the last sentence in the Discussion as follows:

→ **Modified Text (Methods section):** "The score ranges between 0 and 1, with higher values indicating a better match. ICs were removed from the rsfMRI features at the ROI-level using linear regression (Figure 5-A)."

→ **Added Text (Discussion Section):** "The combination of FC and TC measures with age, gender, and TIV could synergistically enhance predictive capacity, with ICs providing additional information that complements these features towards a more robust modeling. In contrast, removing ICs using linear regression may simplify the modeling task by revealing independent individual signals, eliminating potential interactions, and finally, improving identification accuracy. It can also let the predictive model focus on the specific information that FC and TC have collected, leading to more precise identifications."

Comment 10: Finally, I would suggest the authors make the abstract a little clearer and with the main goals clearly

stated (related to 1.). For example, they may want to modify and clarify the subject in the sentence ‘It is also consistent across different levels...’.

Response and action: We appreciate this suggestion and have modified the Abstract as follows:

→ **Modified Abstract:** “In this research, we aimed to compare imaging-derived features of brain function, measured by resting-state fMRI (rsfMRI), with individual characteristics (ICs) such as age, gender, and total intracranial volume (TIV) to predict behavioral measures. Using a dataset of 20,000 healthy individuals from the UK Biobank, we developed a machine learning framework based on rsfMRI features, focusing on temporal complexity (TC) and functional connectivity (FC) measures. Our analysis across four behavioral phenotypes revealed that both TC and FC methods provide comparable predictive performance. However, ICs consistently outperformed rsfMRI features in predictive accuracy, particularly in analyses involving smaller sample sizes. Interestingly, integrating rsfMRI features with demographic data sometimes enhanced predictive outcomes. The efficacy of different predictive modeling techniques and the choice of brain atlas were also examined, showing no significant influence on the results. To summarize, while ICs are superior to rsfMRI in predicting behavioral phenotypes, rsfMRI still conveys additional predictive value in the context of machine learning, such as investigating the roles of specific brain regions or similar objectives.”

Reviewer 2

Comment 1: Are the results dependent on the atlas/regions of interest chosen or are they generalizable beyond the atlas chosen?

Response and action: We acknowledge the significance of this inquiry concerning the applicability of our findings beyond the selection of a brain atlas. In the revised manuscript, we have repeated all analyses in the first study using the Glasser brain atlas with 360 cortical ROIs and using linear SVM in addition to the 400 ROIs of the Schaefer atlas and ridge regression. The results remained stable irrespective of the atlas and predictive modeling used. Specifically, age, gender, and TIV were still better predictors than rsfMRI features across both atlases. This means that our results about how better anatomical and demographic traits can predict behavioural measures are generalizable. The new predictive modeling results have been illustrated in **Figures S1 to S4**. Considering this, we have now added the following paragraph to the Results section:

→ **Added Text (Results Section -- newly added “Robustness analysis” subsection):** “In order to investigate the generalizability of our findings across atlas configurations and predictive models, we performed all analyses using Schaefer400 and Glasser360 brain atlases as well as linear ridge regression and linear SVM. The results of four atlas-model combinations were consistent across various sample sizes and predictive modeling scenarios (see Figure 1). Age, gender, and TIV consistently outperformed rsfMRI features in predicting behavioral phenotypes, irrespective of the atlas and model used. The accuracy curves for predicting behavioral phenotypes over different population sizes using the Schaefer400 atlas were comparable with those obtained using the Glasser360 atlas, for both the ridge regression model (Figure 2 and Figure S1) and the linear SVM model (Figure S2, Figure S3, and Figure S7 to Figure S10). A similar situation was observed when performing age and gender prediction as shown in the pairs of Figure 4 versus Figure S4, Figure S5 versus Figure S6, Figure S11 versus Figure S12, and Figure S13 versus Figure S14. This consistent observation suggests that the superior predictive capacity of demographics and TIV as well as their combinations extend beyond the specific choice of brain atlas and predictive model.

We chose to use ridge regression in our primary study due to its widely recognized application in neuroimaging and behavioral research [Dadi et al., 2019; He et al., 2020]. We then incorporated linear SVM with heuristic hyper parametrization in addition to ridge regression and repeated all analyses using this model. Although both models showed comparable performance at large population sizes, ridge regression demonstrated greater stability, particularly in situations with smaller samples and lower prediction accuracy. This can be observed by comparing the prediction accuracies of visual memory and numeric memory in Figure S2 with their corresponding curves in Figure S3. Furthermore, the linear SVM model necessitates a larger population size to achieve its maximum predictive capability, as opposed to the ridge model. This can be observed by comparing the accuracy curves for processing speed prediction in Figure S2 with the corresponding curves in Figure S3. It justifies the utilization of ridge regression and classification for the behavioral phenotype prediction when using rsfMRI features. A limitation of our study is that we focused exclusively on linear methods for predictive modeling.”

Comment 2: Are the four cognitive phenotypes general enough to represent cognition? Related to this, the authors choose these targets as they are “the four most reliable cognitive phenotypes”. What are the results for unreliable phenotypes? Could they provide a set of results for one unreliable phenotype?

Response and action: Thank you for the question. Examining the reliability of the cognitive and behavioural characteristics in the UK Biobank is a subject of investigation and open inquiry on its own (refer to our most recent study on this matter - Gell et al 2023, available at: <https://www.biorxiv.org/content/10.1101/2023.02.09.527898v2.abstract>). The four behavioural phenotypes selected for this study, namely fluid intelligence, processing speed, visual memory, and numeric memory, were chosen for their reliability within the UK Biobank dataset. It is an important consideration, because unreliable targets can be noisy and lead to less accurate predictions. Therefore, unreliable phenotypes result in low predictive accuracy regardless. We appreciate this interesting comment and consider it for our future research to determine if our results are also applicable to less reliable phenotypes in the UK Biobank database.

Comment 3: What was the motivation for using ridge regression (apart from it being used by other researchers)? It is well known in ML/statistics literature that there exists superior methods to ridge regression. Related to my first comment, are the results dependent on the method chosen or are they generalizable beyond ridge regression?

Response and action: Thank you for the comment. Please also see our response to your **Comment 1**. The selection of a particular predictive model has the potential to bias the results. However, the superiority of one method over another is not known beforehand because it depends on the nature and quality of the input data. Selecting the best model is, thus, an empirically grounded and practically oriented process. Therefore, we used linear ridge regression as our primary method, relying on the existing literature to direct our decision-making. Along with ridge regression, we also used linear SVM with heuristic C in the updated version of our study and ran all the analyses again with this model. The results stayed the same no matter what kind of predictive modeling was used. In other words, age, gender, and TIV were still better predictors than rsfMRI features. We have added a note of this addition to the Methods and Results sections and have acknowledged this in the Discussion section as well to address the concern regarding the choice of prediction method and to highlight the robustness of our findings.

- **Added Text (Discussion Section):** “We chose to use ridge regression in our study due to its widely recognized application in neuroimaging and cognitive research [Dadi et al., 2019; He et al., 2020]. We also incorporated linear SVM with heuristic C in addition to ridge regression and repeated all analyses using this model. Although both models showed comparable performance at large population sizes, ridge regression demonstrated greater stability, particularly in situations with smaller samples and lower prediction accuracy. The use of linear SVM demonstrated limitations, necessitating larger population sizes to achieve optimal predictive capability. A limitation of our study is that we focused exclusively on linear methods for predictive modeling. However, the robustness of the findings over four model-atlas combinations in this study suggests the generalizability of the results.”
- **Added Text (Methods Section):** “In addition to ridge regression, we further explored the robustness of our results by employing linear Support Vector Machines (SVM) with heuristic choice of the hyperparameter C. Linear SVM was chosen as it is a widely used machine learning technique for predictive modeling and can provide an alternative perspective on the relationships between demographic and anatomical factors and behavioural phenotypes. This addition allowed us to assess the robustness of our findings across different regression methods”.
- **Added Text (Results Section):** “Overall, the outcomes achieved using linear SVM closely correspond to those obtained through ridge regression when dealing with large population sizes. However, the ridge model presented less variation in the prediction accuracies when working with smaller sample sizes. Specifically, the ridge models exhibited reduced variability in their prediction accuracy when the accuracy was low. This can be observed by comparing the prediction accuracies of visual memory and numeric memory in Figure S2 with their corresponding curves in Figure S3. Furthermore, the linear SVM model necessitates a larger population size in order to achieve its maximum predictive capability, as opposed to the ridge model. This can be observed by comparing the accuracy curves for processing speed prediction in Figure S2 with the corresponding curves in Figure S3. It justifies the utilization of ridge regression and classification for the behavioural phenotype prediction when using rsfMRI features”.

The linear SVM results have also been added through the new supplementary Figures S1 (using the Schaefer brain atlas), S3 (using the Glasser brain atlas), S5, S6, S8, S10, S12, and S14.

Comment 4: On P.4, the authors state that they used the Spearman correlation between the real and predicted targeted but in the caption in Figure 2 they state Pearson correlation.

Response and action: Thank you for pointing out this inconsistency. We have revised the caption in Figure 2 to accurately state that we used Spearman correlation for evaluating the prediction results.

- **Updated Text (Figure 2):** “Prediction accuracy for behavioural phenotypes and ICs across different sample sizes. The x-axis represents the population size in the analysis, ranging from 100 to 20,000 participants. The y-axis shows the prediction accuracy measured by the Spearman correlation coefficient for behavioural phenotypes (fluid intelligence, processing speed, visual memory, and numeric memory) as well as age, and by the balanced accuracy for gender and fish consumption prediction. The prediction accuracy curves for each behavioural phenotype and individual characteristic are color-coded.”

Comment 5: On P.6 (last line), they authors state Figure 1 – D.1. Where is Figure D.1? D is also mentioned in the first paragraph of P.7.

Response and action: Thank you for bringing this to our attention. Figure D.1 was not included in the manuscript, and its mention was an error. We have removed references to Figure D.1 in both the last line of page 6 and the first paragraph of page 7 to eliminate any confusion.

Comment 6: In the last paragraph of P.7, the authors state that they show “prediction accuracies ... to 60%” in Figure 2. This is not evident. There is also inconsistency in the percentage in the caption of Figure 3 and the figure.

Response and action: Thank you for bringing this to our attention. In the updated Figure 3 as well as Figures S18 to S22 where the 2D prediction accuracy heat maps are illustrated, the y-axis covers the tSNR thresholding from 0% (full set of ROIs) to 100 % (no ROI). We have now made the necessary adjustments to address the issues related to the presentation of prediction accuracies in Figure 3, as well as the supplementary figures. Here are the responses and actions taken:

- **Updated text:** “Figure 3 shows fluid intelligence prediction accuracies for scenarios 1 to 3 after stepwise thresholding on the tSNR maps from 0% (no threshold, equivalent with the results illustrated in Figure 2) to 100% (no ROI for prediction) with 5% increments.”
- **Updated caption:** “Spearman correlation accuracy associated with ridge regression modeling of fluid intelligence using nine rsfMRI features and after tSNR thresholding from 0% to 100%. The numbers of suprathreshold ROIs at tSNR threshold level spanning from 0% to 60% with 5% increment for the Schaefer and Glasser brain atlases are [400, 397, 397, 396, 387, 366, 333, 276, 201, 130, 68, 33, 13] and [360, 360, 356, 353, 338, 311, 270, 203, 142, 83, 57, 25, 10], respectively. For both brain atlases, tSNR levels above 60% led to no suprathreshold ROIs.”

Comment 7: The results in Figure 3 should be explained in more detail in the text.

Response and action: Thank you for your feedback. We have enhanced the explanation of the results shown in Figure 3 by providing a more detailed description in the text. In the Results section, we have expanded the discussion of Figure 3 to offer a clearer understanding of the prediction accuracies for the four behavioural phenotypes. The updates are as follows:

- **Updated text:** “In Figure 3, we present the prediction accuracies for the fluid intelligence target. The figure contains 2D plots depicting Spearman correlation scores obtained from ridge regression modeling of fluid intelligence using nine rsfMRI features, subject to tSNR thresholding ranging from 0% to 60%. Each panel in the figure represents a distinct pair of features and targets, with color-coded accuracy values. The x-axis indicates the population size in the analysis, while the y-axis denotes the count of suprathreshold ROIs after tSNR thresholding. The predictive modeling for each feature-target pair is conducted across various sample sizes, spanning $N_{\text{subject}} = 100$ to $N_{\text{subject}} = 20,000$. For population sizes between 100 and 2000, increments of 50 subjects were employed, while increments of 500 subjects were applied for the range of 2000 to 20,000.”

Comment 8: Why are the results from predicting TIV not included in Figure 4?

Response and action: Thank you for this relevant question. The incorporation of age and gender was primarily based on their well-established reliability as predictive factors in the literature. However, this is not the case for TIV to the best of our knowledge, making it a novel conceptual direction. Therefore, we consider this avenue of research for our future work.

Comment 9: What is the definition of balanced accuracy?

Response and action: Balanced accuracy is a performance measure utilized in classification tasks that considers the imbalances in the distribution of classes. The term is defined as the mathematical average of sensitivity and specificity, where sensitivity is the ratio of correctly identified positive instances, and specificity is the ratio of correctly identified negative instances by the classifier.

Comment 10: In the Discussion, the authors consider the problem of dataset decay. But this would only be applicable to inference or testing, which is not the focus of this paper (prediction).

Response and action: Thank you for your comment. Considering this feedback, we chose to exclude the concept of dataset decay from the revised manuscript. The following part was removed from the Discussion:

- **Original Text (Discussion Section):** “In addition, it is important to consider the potential issue of dataset decay [ref]. As neuroimaging datasets age, the relevance and representativeness of the data may decrease, potentially impacting the generalizability of predictive models. This phenomenon can be particularly relevant in longitudinal

studies or when attempting to apply models trained on older data to new populations. Therefore, future work should explore strategies to mitigate the effects of dataset decay and maintain the robustness of predictive models over time."

Comment 11: The grid search for the hyperparameter λ should be increased to cover more values.

Response and action: Thank you for your suggestion. We have expanded the grid search for the hyperparameter of the ridge models to cover a much wider range of values than before, i.e., [0, 0.00001, 0.0001, 0.001, 0.004, 0.007, 0.01, 0.04, 0.07, 0.1, 0.4, 0.7, 1, 1.5, 2, 2.5, 3, 3.5, 4, 5, 10, 15, 20, 30, 40, 50, 60, 70, 80, 100, 150, 200, 300, 500, 700, 1000, 10,000, 100,000, 1,000,000]. This extended search allowed us to explore a more comprehensive range of parameter values and select the optimal one for our analysis (see Figure S15). Although it helped the prediction accuracies to some extent, it did not change the main findings of our study significantly and the model performance metrics and the relationships between different features remained consistent with the previous findings.

References

- Brown M, Sidhu G, Greiner R, Asgarian N, Bastani M, Silverstone P, Greenshaw A, Dursun S (2012): ADHD-200 Global Competition: diagnosing ADHD using personal characteristic data can outperform resting state fMRI measurements. *Front Syst Neurosci* 6. <https://www.frontiersin.org/articles/10.3389/fnsys.2012.00069>.
- Cui Z, Gong G (2018): The effect of machine learning regression algorithms and sample size on individualized behavioral prediction with functional connectivity features. *NeuroImage* 178:622–637.
- Dadi K, Rahim M, Abraham A, Chyzyk D, Milham M, Thirion B, Varoquaux G, Alzheimer's Disease Neuroimaging Initiative (2019): Benchmarking functional connectome-based predictive models for resting-state fMRI. *NeuroImage* 192:115–134.
- Dadi K, Varoquaux G, Houenou J, Bzdok D, Thirion B, Engemann D (2021): Population modeling with machine learning can enhance measures of mental health. *GigaScience* 10:giab071.
- Glasser MF, Coalson TS, Robinson EC, Hacker CD, Harwell J, Yacoub E, Ugurbil K, Andersson J, Beckmann CF, Jenkinson M, Smith SM, Van Essen DC (2016): A multi-modal parcellation of human cerebral cortex. *Nature* 536:171–178.
- He T, Kong R, Holmes AJ, Nguyen M, Sabuncu MR, Eickhoff SB, Bzdok D, Feng J, Yeo BTT (2020): Deep neural networks and kernel regression achieve comparable accuracies for functional connectivity prediction of behavior and demographics. *NeuroImage* 206:116276.
- Liégeois R, Li J, Kong R, Orban C, Van De Ville D, Ge T, Sabuncu MR, Yeo BTT (2019): Resting brain dynamics at different timescales capture distinct aspects of human behavior. *Nat Commun* 10:2317.
- Michon KJ, Khammash D, Simmonite M, Hamlin AM, Polk TA (2022): Person-specific and precision neuroimaging: Current methods and future directions. *NeuroImage* 263:119589.
- Miller KL, Alfaro-Almagro F, Bangerter NK, Thomas DL, Yacoub E, Xu J, Bartsch AJ, Jbabdi S, Sotiropoulos SN, Andersson JLR, Griffanti L, Douaud G, Okell TW, Weale P, Dragonu I, Garratt S, Hudson S, Collins R, Jenkinson M, Matthews PM, Smith SM (2016): Multimodal population brain imaging in the UK Biobank prospective epidemiological study. *Nat Neurosci* 19:1523–1536.
- Murphy K, Bodurka J, Bandettini PA (2007): How long to scan? The relationship between fMRI temporal signal to noise and necessary scan duration. *NeuroImage* 34:565–574.
- Schaefer A, Kong R, Gordon EM, Laumann TO, Zuo X-N, Holmes AJ, Eickhoff SB, Yeo BTT (2018): Local-Global Parcellation of the Human Cerebral Cortex from Intrinsic Functional Connectivity MRI. *Cereb Cortex N Y N* 1991 28:3095–3114.
- Tejavibulya L, Rolison M, Gao S, Liang Q, Peterson H, Dadashkarimi J, Farruggia MC, Hahn CA, Noble S, Lichenstein SD, Pollatou A, Dufford AJ, Scheinost D (2022): Predicting the future of neuroimaging predictive models in mental health. *Mol Psychiatry* 27:3129–3137.

Figures

Figure 1: (A): main block diagram of this study, including the rsfMRI features and the prediction targets from the UK Biobank. (B) Four analysis scenarios based on the role of individual characteristics, i.e., age, gender, and total intracranial volume (TIV), in cognitive phenotypic prediction.

Figure 2: Prediction accuracy for behavioural phenotypes and ICs across different sample sizes. The x-axis represents the population size in the analysis, ranging from 100 to 20,000 participants. The y-axis shows the prediction accuracy measured by the Spearman correlation coefficient for behavioural phenotypes (fluid intelligence, processing speed, visual memory, and numeric memory) as well as age, and by the balanced accuracy for gender and fish consumption prediction. The prediction accuracy curves for each behavioural phenotype and individual characteristic are color-coded.

Fluid Intelligence Prediction

(A) Scenario 1: Prediction using rsfMRI features without removing individual characteristics

(B) Scenario 2: Prediction using rsfMRI features with removing individual characteristics

(C) Scenario 3: Prediction using combined rsfMRI features and individual characteristics

Figure 3: Spearman correlation accuracy associated with ridge regression modeling of fluid intelligence using nine rsfMRI features and after tSNR thresholding from 0% to 100%. The numbers of suprathreshold ROIs at tSNR threshold level spanning from 0% to 60% with 5% increment for the Schaefer and Glasser brain atlases are [400, 397, 397, 396, 387, 366, 333, 276, 201, 130, 68, 33, 13] and [360, 360, 356, 353, 338, 311, 270, 203, 142, 83, 57, 25, 10], respectively. For both brain atlases, tSNR levels above 60% led to no suprathreshold ROIs.

Figure 4: Prediction accuracy scores associated with nine rsfMRI features and age and gender as targets using scenarios 1–3 of this study (see also Figures 1-B.1–B.3 and Methods). The prediction accuracies of individual characteristics only (Scenario 4 in Figure 1-B.4) have been plotted in bold black on all panels. For age prediction, we considered gender and TIV as confounds, while for gender prediction, we considered age and TIV as confounds. Age prediction accuracies are computed as the Pearson correlation between the actual values and predicted values through ridge regression modeling. Gender prediction accuracies are computed as the balanced accuracy through ridge binary classification. Each rsfMRI feature is illustrated in a distinct color and listed in the figure legend. In each figure panel, the x-axis represents the population size in the analysis, and the y-axis shows the prediction accuracy. The predictive modeling of each pair of features and targets is repeated for different sample sizes in the UK Biobank, ranging from $N_{subject} = 100$ to $N_{subject} = 20,000$. The population sizes from 100 to 2000 were increased with a 50-step increment (see the light orange shadow in the figure panels) and from 2000 to 20,000 with a 500-step increment (see the light blue shadow in the figure panels). See Supplementary Figure S2 for the boxplot representation of these results.

Figure 5: The process and results of rsfMRI feature comparison. (A) A schematic example of comparing two rsfMRI features X and Y from the same subject in a sample. This comparison leads to the computation of an identification accuracy score (see Methods). (B-D) Identification accuracy patterns of 10 rsfMRI feature pairs with above zero matching are associated with three analysis scenarios of this study (see Figure 1 as well as Methods). Each pair in the middle row panels has been depicted in a distinct color, and all pairs are listed in the figure legend. In each figure panel, the x-axis represents the population size in the analysis, and the y-axis shows the identification accuracy. The identification analyses are repeated for different sample sizes in the UK Biobank, ranging from $N_{subject} = 100$ to $N_{subject} = 20,000$. The population sizes from 100 to 2000 were increased with a 50-step increment (see the light orange shadow in the figure panels) and from 2000 to 20,000 with a 500-step increment (see the light blue shadow in the figure panels). The color-coded matrices in the row illustrate the identification accuracy of rsfMRI feature pairs for $N_{subject} = 20,000$.

Figure S 1: Prediction accuracy scores associated with nine rsfMRI features and five prediction targets using scenarios 1–3 of this study using the Schaefer brain atlas and linear SVM predictive modeling (see also Figures 1-B.1–B.3 and Methods). The prediction accuracies of individual characteristics only (Scenario 4 in Figure 1-B.4) have been plotted in bold black on all panels. Prediction accuracies of the fluid intelligence, processing speed, visual memory, and numeric memory scores are computed as the Spearman correlation between the actual values and predicted values through SVM modeling. The prediction accuracy of Fish consumer yesterday is computed as the balanced accuracy through SVM binary classification. Each rsfMRI feature is illustrated in a distinct color and listed in the figure legend. In each figure panel, the x-axis represents the population size in the analysis, and the y-axis shows the prediction accuracy. The predictive modeling of each pair of features and targets is repeated for different sample sizes in the UK Biobank, ranging from $N_{\text{subject}} = 100$ to $N_{\text{subject}} = 20,000$.

The population sizes from 100 to 2000 were increased with a 50-step increment and from 2000 to 20,000 with a 500-step increment.

See Figure S8 for the boxplot representation of these results.

Figure S 2: Prediction accuracy scores associated with nine rsfMRI features and five prediction targets using scenarios 1–3 of this study using the Glasser brain atlas and ridge predictive modeling (see also Figures 1-B.1–B.3 and Methods). The prediction accuracies of individual characteristics only (Scenario 4 in Figure 1-B.4) have been plotted in bold black on all panels. Prediction accuracies of the fluid intelligence, processing speed, visual memory, and numeric memory scores are computed as the Spearman correlation between the actual values and predicted values through ridge regression modeling. The prediction accuracy of Fish consumer yesterday is computed as the balanced accuracy through ridge binary classification. Each rsfMRI feature is illustrated in a distinct color and listed in the figure legend. In each figure panel, the x-axis represents the population size in the analysis, and the y-axis shows the prediction accuracy. The predictive modeling of each pair of features and targets is repeated for different sample sizes in the UK Biobank, ranging from $N_{\text{subject}} = 100$ to $N_{\text{subject}} = 20,000$. The population sizes from 100 to 2000 were increased with a 50-step increment and from 2000 to 20,000 with a 500-step increment. See Figure S9 for the boxplot representation of these results.

Figure S 3: Prediction accuracy scores associated with nine rsfMRI features and five prediction targets using scenarios 1–3 of this study using the Glasser brain atlas and linear SVM predictive modeling (see also Figures 1-B.1–B.3 and Methods). The prediction accuracies of individual characteristics only (Scenario 4 in Figure 1-B.4) have been plotted in bold black on all panels. Prediction accuracies of the fluid intelligence, processing speed, visual memory, and numeric memory scores are computed as the Spearman correlation between the actual values and predicted values through SVM modeling. The prediction accuracy of Fish consumer yesterday is computed as the balanced accuracy through SVM binary classification. Each rsfMRI feature is illustrated in a distinct color and listed in the figure legend. In each figure panel, the x-axis represents the population size in the analysis, and the y-axis shows the prediction accuracy. The predictive modeling of each pair of features and targets is repeated for different sample sizes in the UK Biobank, ranging from $N_{\text{subject}} = 100$ to $N_{\text{subject}} = 20,000$.

The population sizes from 100 to 2000 were increased with a 50-step increment and from 2000 to 20,000 with a 500-step increment.

See Figure S10 for the boxplot representation of these results.

Figure S 4: Prediction accuracy scores associated with nine rsfMRI features and age and gender as targets using scenarios 1–3 of this study using the Schaefer brain atlas and linear SVM predictive modeling (see also Figures 1-B.1–B.3 and Methods). For age prediction, we considered gender and TIV as confounds, while for gender prediction, we considered age and TIV as confounds. Age prediction accuracies are computed as the Pearson correlation between the actual values and predicted values through SVM modeling. Gender prediction accuracies are computed as the balanced accuracy through SVM binary classification. Each rsfMRI feature is illustrated in a distinct color and listed in the figure legend. The population sizes from 100 to 2000 were increased with a 50-step increment and from 2000 to 20,000 with a 500-step increment. See Figure S12 for the boxplot representation of these results.

Figure S 5: Prediction accuracy scores associated with nine rsfMRI features and age and gender as targets using scenarios 1–3 of this study using the Glasser brain atlas and ridge predictive modeling (see also Figures 1-B.1–B.3 and Methods). For age prediction, we considered gender and TIV as confounds, while for gender prediction, we considered age and TIV as confounds. Age prediction accuracies are computed as the Pearson correlation between the actual values and predicted values through ridge regression modeling. Gender prediction accuracies are computed as the balanced accuracy through ridge binary classification. Each rsfMRI feature is illustrated in a distinct color and listed in the figure legend. The population sizes from 100 to 2000 were increased with a 50-step increment and from 2000 to 20,000 with a 500-step increment. See Figure S13 for the boxplot representation of these results.

Figure S 6: Prediction accuracy scores associated with nine rsfMRI features and age and gender as targets using scenarios 1–3 of this study using the Glasser brain atlas and linear SVM predictive modeling (see also Figures 1-B.1–B.3 and Methods). For age prediction, we considered gender and TIV as confounds, while for gender prediction, we considered age and TIV as confounds. Age prediction accuracies are computed as the Pearson correlation between the actual values and predicted values through SVM modeling. Gender prediction accuracies are computed as the balanced accuracy through SVM binary classification. Each rsfMRI feature is illustrated in a distinct color and listed in the figure legend. The population sizes from 100 to 2000 were increased with a 50-step increment and from 2000 to 20,000 with a 500-step increment. See Figure S14 for the boxplot representation of these results.

(A) Scenario 1: Prediction using rsfMRI features without removing individual characteristics

(B) Scenario 2: Prediction using rsfMRI features with removing individual characteristics

(C) Scenario 3: Prediction using combined rsfMRI features and individual characteristics

Figure S 7: Prediction accuracy scores associated with nine rsfMRI features and five prediction targets using scenarios 1–3 of this study using the Schaefer brain atlas and ridge predictive modeling (see also Figures 1-B.1–B.3 and Methods). Prediction accuracies of the fluid intelligence, processing speed, visual memory, and numeric memory scores are computed as the Pearson correlation between the actual values and predicted values through ridge regression modeling. The prediction accuracy of Fish consumer yesterday is computed as the balanced accuracy through ridge binary classification. Each rsfMRI feature is illustrated in a distinct color and listed in the figure legend. In each figure panel, the box has a line at the median and spans the complete range of sample sizes (from 100 to 20,000 participants), extending from the lower to upper quartile values of the prediction accuracies. The whiskers extend outside the box to display the data's range. The population sizes from 100 to 2000 were increased with a 50-step increment and from 2000 to 20,000 with a 500-step increment. See Figure 2 for the representation of prediction accuracies over the range of sample sizes.

Figure S 8: Prediction accuracy scores associated with nine rsfMRI features and five prediction targets using scenarios 1–3 of this study using the Schaefer brain atlas and linear SVM predictive modeling (see also Figures 1-B.1–B.3 and Methods). Prediction accuracies of the fluid intelligence, processing speed, visual memory, and numeric memory scores are computed as the Pearson correlation between the actual values and predicted values through SVM modeling. The prediction accuracy of Fish consumer yesterday is computed as the balanced accuracy through SVM binary classification. Each rsfMRI feature is illustrated in a distinct color and listed in the figure legend. In each figure panel, the box has a line at the median and spans the complete range of sample sizes (from 100 to 20,000 participants), extending from the lower to upper quartile values of the prediction accuracies. The whiskers extend outside the box to display the data's range. The population sizes from 100 to 2000 were increased with a 50-step increment and from 2000 to 20,000 with a 500-step increment. See Figure S1 for the representation of prediction accuracies over the range of sample sizes.

Figure S 9: Prediction accuracy scores associated with nine rsfMRI features and five prediction targets using scenarios 1–3 of this study using the Glasser brain atlas and ridge predictive modeling (see also Figures 1-B.1–B.3 and Methods). Prediction accuracies of the fluid intelligence, processing speed, visual memory, and numeric memory scores are computed as the Pearson correlation between the actual values and predicted values through ridge regression modeling. The prediction accuracy of Fish consumer yesterday is computed as the balanced accuracy through ridge binary classification. Each rsfMRI feature is illustrated in a distinct color and listed in the figure legend. In each figure panel, the box has a line at the median and spans the complete range of sample sizes (from 100 to 20,000 participants), extending from the lower to upper quartile values of the prediction accuracies. The whiskers extend outside the box to display the data's range. The population sizes from 100 to 2000 were increased with a 50-step increment and from 2000 to 20,000 with a 500-step increment. See Figure S2 for the representation of prediction accuracies over the range of sample sizes.

(A) Scenario 1: Prediction using rsfMRI features without removing individual characteristics

(B) Scenario 2: Prediction using rsfMRI features with removing individual characteristics

(C) Scenario 3: Prediction using combined rsfMRI features and individual characteristics

Figure S 10: Prediction accuracy scores associated with nine rsfMRI features and five prediction targets using scenarios 1–3 of this study using the Glasser brain atlas and linear SVM predictive modeling (see also Figures 1-B.1–B.3 and Methods). Prediction accuracies of the fluid intelligence, processing speed, visual memory, and numeric memory scores are computed as the Pearson correlation between the actual values and predicted values through SVM modeling. The prediction accuracy of Fish consumer yesterday is computed as the balanced accuracy through SVM binary classification. Each rsfMRI feature is illustrated in a distinct color and listed in the figure legend. In each figure panel, the box has a line at the median and spans the complete range of sample sizes (from 100 to 20,000 participants), extending from the lower to upper quartile values of the prediction accuracies. The whiskers extend outside the box to display the data’s range. The population sizes from 100 to 2000 were increased with a 50-step increment and from 2000 to 20,000 with a 500-step increment. See Figure S3 for the representation of prediction accuracies over the range of sample sizes.

Figure S 11: Prediction accuracy scores associated with nine rsfMRI features and age and gender as targets using scenarios 1–3 of this study using the Schaefer brain atlas and ridge predictive modeling (see also Figures 1-B.1–B.3 and Methods). For age prediction, we considered gender and TIV as confounds, while for gender prediction, we considered age and TIV as confounds. Age prediction accuracies are computed as the Pearson correlation between the actual values and predicted values through ridge regression modeling. Gender prediction accuracies are computed as the balanced accuracy through ridge binary classification. Each rsfMRI feature is illustrated in a distinct color and listed in the figure legend. In each figure panel, the box has a line at the median and spans the complete range of sample sizes (from 100 to 20,000 participants), extending from the lower to upper quartile values of the prediction accuracies. The whiskers extend outside the box to display the data’s range. The population sizes from 100 to 2000 were increased with a 50-step increment and from 2000 to 20,000 with a 500-step increment. See Figure 4 for the representation of prediction accuracies over the range of sample sizes.

Figure S 12: Prediction accuracy scores associated with nine rsfMRI features and age and gender as targets using scenarios 1–3 of this study using the Schaefer brain atlas and linear SVM predictive modeling (see also Figures 1-B.1–B.3 and Methods). For age prediction, we considered gender and TIV as confounds, while for gender prediction, we considered age and TIV as confounds. Age prediction accuracies are computed as the Pearson correlation between the actual values and predicted values through SVM modeling. Gender prediction accuracies are computed as the balanced accuracy through SVM binary classification. Each rsfMRI feature is illustrated in a distinct color and listed in the figure legend. In each figure panel, the box has a line at the median and spans the complete range of sample sizes (from 100 to 20,000 participants), extending from the lower to upper quartile values of the prediction accuracies. The whiskers extend outside the box to display the data’s range. The population sizes from 100 to 2000 were increased with a 50-step increment and from 2000 to 20,000 with a 500-step increment. See Figure S4 for the representation of prediction accuracies over the range of sample sizes.

Figure S 13: Prediction accuracy scores associated with nine rsfMRI features and age and gender as targets using scenarios 1–3 of this study using the Glasser brain atlas and ridge predictive modeling (see also Figures 1-B.1–B.3 and Methods). For age prediction, we considered gender and TIV as confounds, while for gender prediction, we considered age and TIV as confounds. Age prediction accuracies are computed as the Pearson correlation between the actual values and predicted values through ridge regression modeling. Gender prediction accuracies are computed as the balanced accuracy through ridge binary classification. Each rsfMRI feature is illustrated in a distinct color and listed in the figure legend. In each figure panel, the box has a line at the median and spans the complete range of sample sizes (from 100 to 20,000 participants), extending from the lower to upper quartile values of the prediction accuracies. The whiskers extend outside the box to display the data’s range. The population sizes from 100 to 2000 were increased with a 50-step increment and from 2000 to 20,000 with a 500-step increment. See Figure S5 for the representation of prediction accuracies over the range of sample sizes.

Figure S 14: Prediction accuracy scores associated with nine rsfMRI features and age and gender as targets using scenarios 1–3 of this study using the Glasser brain atlas and linear SVM predictive modeling (see also Figures 1-B.1–B.3 and Methods). For age prediction, we considered gender and TIV as confounds, while for gender prediction, we considered age and TIV as confounds. Age prediction accuracies are computed as the Pearson correlation between the actual values and predicted values through SVM modeling. Gender prediction accuracies are computed as the balanced accuracy through SVM binary classification. Each rsfMRI feature is illustrated in a distinct color and listed in the figure legend. In each figure panel, the box has a line at the median and spans the complete range of sample sizes (from 100 to 20,000 participants), extending from the lower to upper quartile values of the prediction accuracies. The whiskers extend outside the box to display the data’s range. The population sizes from 100 to 2000 were increased with a 50-step increment and from 2000 to 20,000 with a 500-step increment. See Figure S6 for the representation of prediction accuracies over the range of sample sizes.

(A) Schaefer brain atlas

(B) Glasser brain atlas

Figure S 15: Optimal alpha parameter of the ridge models in four confound removal scenarios for rsfMRI feature vectors generated by (A) Schaefer brain atlas with $N_{ROI}=400$, and (B) Glasser brain atlas with $N_{ROI}=360$. The distributions are associated with the rsfMRI feature vectors at no tSNR thresholding, i.e., utilizing complete number of ROIs in each brain atlas.

Figure S 16: Prediction accuracy scores associated with nine rsfMRI features and five prediction targets using scenarios 1–3 of this study for the population size of $N_{\text{subject}} = 20,000$ using the Schaefer brain atlas and linear SVM predictive modeling (see also Figures 1-B.1–B.3 and Methods). Prediction accuracies of the fluid intelligence, processing speed, visual memory, and numeric memory scores are computed as the Spearman correlation between the actual values and predicted values through SVM modeling. The prediction accuracy of Fish consumer yesterday is computed as the balanced accuracy through SVM binary classification. Each rsfMRI feature is illustrated in a distinct color and listed in the figure legend. In each figure panel, the x-axis represents the number of suprathreshold ROIs after tSNR thresholding from 0% to 100%, and the y-axis shows the prediction accuracy.

Figure S 17: Prediction accuracy scores associated with nine rsfMRI features and five prediction targets using scenarios 1–3 of this study for the population size of $N_{\text{subject}} = 20,000$ using the Schaefer brain atlas and linear SVM predictive modeling (see also Figures 1-B.1–B.3 and Methods). Prediction accuracies of the fluid intelligence, processing speed, visual memory, and numeric memory scores are computed as the Spearman correlation between the actual values and predicted values through SVM modeling. The prediction accuracy of Fish consumer yesterday is computed as the balanced accuracy through SVM binary classification. Each rsfMRI feature is illustrated in a distinct color and listed in the figure legend. In each figure panel, the x-axis represents the tSNR thresholding levels, applied on the rsfMRI feature vectors, from 0% to 100%, and the y-axis shows the prediction accuracy.

Processing Speed Prediction

(A) Scenario 1: Prediction using rsfMRI features without removing individual characteristics

(B) Scenario 2: Prediction using rsfMRI features with removing individual characteristics

(C) Scenario 3: Prediction using combined rsfMRI features and individual characteristics

Figure S 18: Spearman correlations associated with the Schaefer brain atlas and ridge regression modeling of the processing speed score using nine rsfMRI features and after tSNR thresholding from 0% (no threshold) to 100%. In each figure panel, the accuracy values are color-coded. Additionally, the x-axis represents the population size in the analysis, and the y-axis shows the number of suprathreshold ROIs after tSNR thresholding. The predictive modeling of each pair of features and targets is repeated for different sample sizes in the UK Biobank, ranging from $N_{\text{subject}} = 100$ to $N_{\text{subject}} = 20,000$. The population sizes from 100 to 2000 were increased with a 50-step increment, and from 2000 to 20,000 with a 500-step increment.

Numeric Memory Prediction

(A) Scenario 1: Prediction using rsfMRI features without removing individual characteristics

(B) Scenario 2: Prediction using rsfMRI features with removing individual characteristics

(C) Scenario 3: Prediction using combined rsfMRI features and individual characteristics

Figure S 19: Spearman correlations associated with the Schaefer brain atlas and ridge regression modeling of the numeric memory score using nine rsfMRI features and after tSNR thresholding from 0% (no threshold) to 65%. In each figure panel, the accuracy values are color-coded. Additionally, the x-axis represents the population size in the analysis, and the y-axis shows the number of suprathreshold ROIs after tSNR thresholding. The predictive modeling of each pair of features and targets is repeated for different sample sizes in the UK Biobank, ranging from $N_{\text{subject}} = 100$ to $N_{\text{subject}} = 20,000$. The population sizes from 100 to 2000 were increased with a 50-step increment, and from 2000 to 20,000 with a 500-step increment.

Visual Memory Prediction

Figure S 20: Spearman correlations associated with the Schaefer brain atlas and ridge regression modeling of the visual memory score using nine rsfMRI features and after tSNR thresholding from 0% (no threshold) to 65%. In each figure panel, the accuracy values are color-coded. Additionally, the x-axis represents the population size in the analysis, and the y-axis shows the number of suprathreshold ROIs after tSNR thresholding. The predictive modeling of each pair of features and targets is repeated for different sample sizes in the UK Biobank, ranging from Nsubject = 100 to Nsubject = 20,000. The population sizes from 100 to 2000 were increased with a 50-step increment, and from 2000 to 20,000 with a 500-step increment.

Fish Consumption Prediction

Figure S 21: Spearman correlations associated with the Schaefer brain atlas and ridge regression modeling of the Fish consumption yesterday using nine rsfMRI features and after tSNR thresholding from 0% (no threshold) to 65%. In each figure panel, the accuracy values are color-coded. Additionally, the x-axis represents the population size in the analysis, and the y-axis shows the number of suprathreshold ROIs after tSNR thresholding. The predictive modeling of each pair of features and targets is repeated for different sample sizes in the UK Biobank, ranging from Nsubject = 100 to Nsubject = 20,000. The population sizes from 100 to 2000 were increased with a 50-step increment, and from 2000 to 20,000 with a 500-step increment.

Figure S 22: Group level mean tSNR brain maps obtained using the (A) Schaefer and (B) Glasser brain atlases. The maps have been averaged after min-max normalization of the subject-specific tSNR maps over the entire population of 20,000 UK Biobank subjects. The numbers of suprathreshold ROIs at tSNR threshold level spanning from 0% to 60% with 5% increment for the Schaefer and Glasser brain atlases are [400, 397, 397, 396, 387, 366, 333, 276, 201, 130, 68, 33, 13] and [360, 360, 356, 353, 338, 311, 270, 203, 142, 83, 57, 25, 10], respectively. For both brain atlases, tSNR levels above 60% led to no suprathreshold ROIs.

REVIEWERS' COMMENTS:

Reviewer #1 (Remarks to the Author):

The authors addressed all of my comments and concerns very well and I would like to congratulate them on a nice manuscript.

A minor comment - I appreciate that the authors included another parcellation, however, it would be helpful if they clearly indicate in captions of Figures 2 and 3, and maybe in the text as well, which parcellation was used.